# A Survey on Deep Learning Approaches for Tabular Data Generation: Utility, Alignment, Fidelity, Privacy, Diversity, and Beyond

**Mihaela Cătălina Stoian**                                      *m.stoian@imperial.ac.uk*
*University of Oxford, Imperial College London*

**Eleonora Giunchiglia**                                      *e.giunchiglia@imperial.ac.uk*
*Imperial College London*

**Thomas Lukasiewicz**                                      *thomas.lukasiewicz@tuwien.ac.at*
*Vienna University of Technology, University of Oxford*

**Reviewed on OpenReview:** *https://openreview.net/forum?id=RoShSRQQ67*

## Abstract

Generative modelling has become the standard approach for synthesising tabular data. However, different use cases demand synthetic data to comply with different requirements to be useful in practice. In this survey, we review deep generative modelling approaches for tabular data from the perspective of five types of requirements: utility of the synthetic data, alignment of the synthetic data with domain-specific knowledge, statistical fidelity of the synthetic data distribution compared to the real data distribution, privacy-preserving capabilities, and sampling diversity. We group the approaches along two levels of granularity: (i) based on the requirements they address and (ii) according to the underlying model they utilise. Additionally, we summarise the appropriate evaluation methods for each requirement, the relationships among the requirements, and the specific characteristics of each model type. Finally, we discuss future directions for the field, along with opportunities to improve the current evaluation methods. Overall, this survey can be seen as a user guide to tabular data generation: helping readers navigate available models and evaluation methods to find those best suited to their needs.

## 1 Introduction

In recent years, the synthesis of realistic tabular data has emerged as a critical research area, driven by the need for privacy-preserving machine learning, robust AI benchmarking, and data augmentation in high-stakes applications such as finance and healthcare. The successful deployment of deep generative modelling led to synthesisers with the ability to improve utility in downstream tasks (e.g., [Kim et al., 2023]), reduce data scarcity by augmenting existing datasets (e.g., [Jia et al., 2024]), and create new datasets in scenarios where privacy is an important concern (e.g., [Vero et al., 2024]) and anonymising data is a challenging task.

Yet, synthesising such data comes with unique challenges that distinguish it from unstructured domains like computer vision or natural language. Indeed, differently from images or text, where data is homogeneous and presents spatial or sequential correlations, tabular data is inherently heterogeneous, comprising a complex mix of continuous, discrete, and categorical features with diverse distributions. Further, the relations between these features are often non-linear and lack the local structures and patterns leveraged by standard deep learning architectures such as convolutional neural networks. Hence, synthesisers for tabular data face the complex task of faithfully reconstructing these correlations while also handling inherent irregularities such as data imbalance. Moreover, once the data is generated, evaluating its quality proves

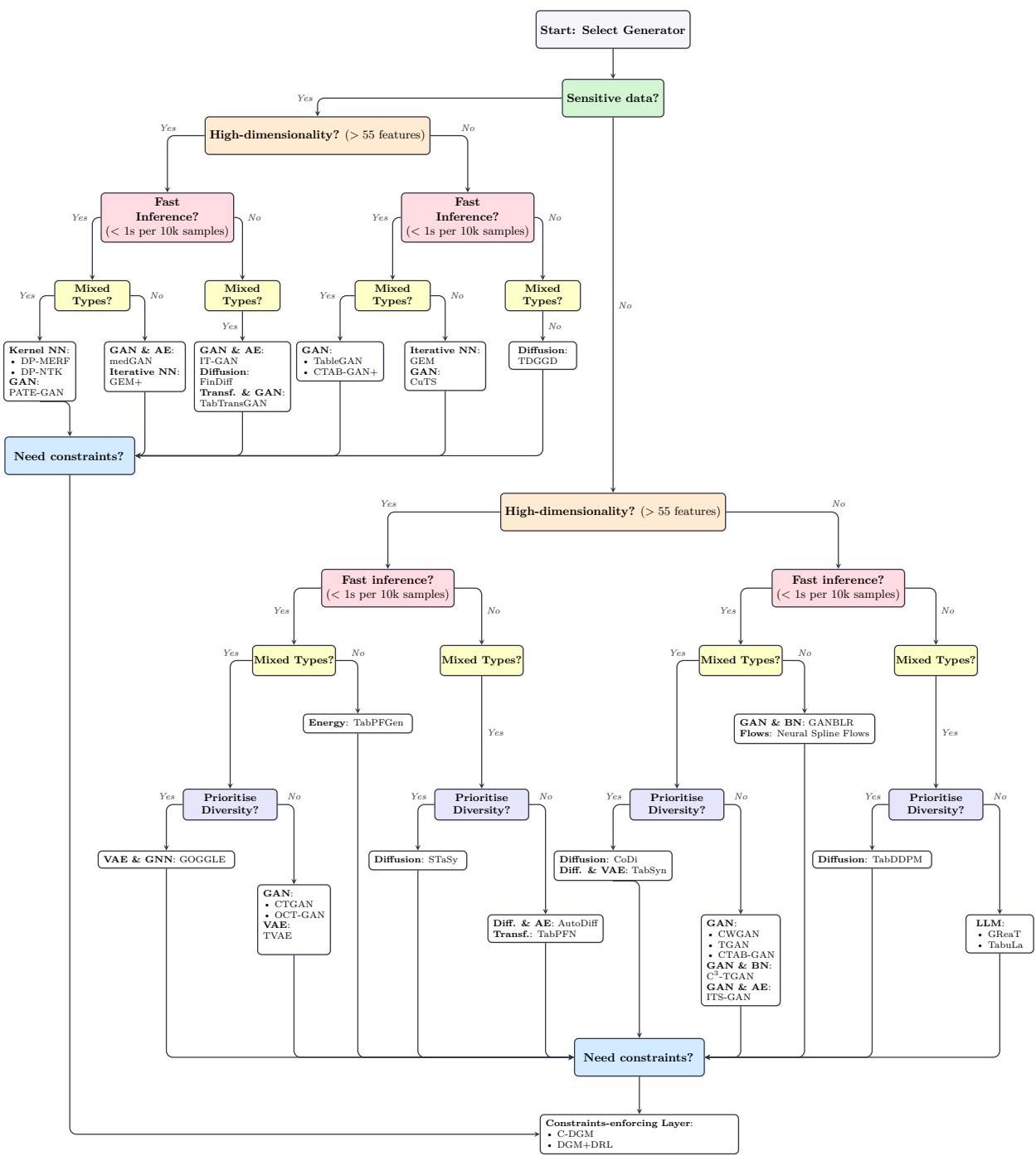

Figure 1: Simplified roadmap to help select an appropriate tabular data synthesiser based on specific requirements, as well as other considerations such as whether the data is high-dimensional and whether the sampling time needs to be low. Note that in this roadmap, we consider fast the models that take less than 1 second to generate 10k samples.

challenging [Naeem et al., 2020; Räisä et al., 2025], particularly regarding whether statistical inferences can be faithfully reproduced [Montoya Perez et al., 2024], as there is no single metric capable of capturing all the desired requirements.

These diverse needs and technical challenges have led to not only a proliferation of different generative models, but also to an explosion of diverse evaluation metrics whose aim is to evaluate whether a model is able to address all the requirements of the given use case. However, the sheer diversity of these metrics has made it increasingly difficult to assess and compare different synthesisers in a standardised manner.

This survey analyses models and evaluations methods alike from the point of view of the user, categorising them on the ground of the requirements they address and measure: utility, alignment, fidelity, privacy, and diversity. The analysis thus serves as a structured resource for both newcomers and experienced practitioners seeking a formal evaluation framework for tabular data synthesis. To this end, we first give an overview of the tabular data synthesis problem and its requirements (§2). Then, we detail each requirement along with its evaluation protocols and metrics (§3), but also the relationships among these requirements (§3.6). Next, we survey the existing deep learning generative models for synthesising tabular data (§4), grouping them by their model architecture, but also by the requirements they address, as shown in Table 1 and in Figure 3. In particular, Table 1 can be seen as a summary of the surveyed approaches (ordered by their date of publication), providing also an overview of the characteristics for each model, such as their ability to handle high-dimensional and mixed data types, as well as their sample generation times. We lastly review other deep generative models originally developed for other fields (such as natural language processing and computer vision), which can be adapted to tabular data synthesis (§4.6). We conclude with a discussion on related surveys and future directions and opportunities for improvement (§5).

Crucially, throughout this survey, we analyse the key requirements that practitioners typically face when synthesising data. Building on this analysis, in Figure 1, we present a roadmap designed to guide readers in selecting the appropriate synthesiser for their specific use case. This roadmap thus serves as a simplified visualisation of the surveyed methods, with methods' grouping based on critical considerations such as data sensitivity, high-dimensionality, sampling speed, but also the need for the generated data to adhere to domain constraints. Notably, the decision to enforce domain constraints is positioned as a final step. Indeed, in case constraints are needed, the listed methods to guarantee their satisfaction can be simply built on top of a given synthesiser as they are differentiable layers able to be integrated into any neural network model during training. Conversely, the decision regarding privacy is fundamental and, thus, appears as the first step in the roadmap, specifying whether a specialised privacy-preserving mechanism is required from the start.

## 2 Synthesising Tabular Data

**Tabular Dataset.** In the tabular data synthesis field, a tabular dataset $\mathcal{D}$ is a collection of $N$ i.i.d. samples (indexed by $i \in \{1, \ldots, N\}$ such that the $i$-th sample constitutes the $i$-th row) drawn from an unknown joint distribution $p_{\mathcal{X}}$ over the variables (i.e., columns) $(X_1, X_2, \ldots, X_K)$. Thus, for each $i \in \{1, \ldots, N\}$, the sample $\mathbf{x}^i = (x_1^i, x_2^i, \ldots, x_K^i)$ is an assignment of values to the variables $(X_1, X_2, \ldots, X_K)$, i.e., $x_j^i \in \mathbb{D}^j$ is the value assigned to variable $X_j$, for each $j \in \{1, \ldots, K\}$, where $\mathbb{D}^j$ is the set of permissible values for variable $X_j$. One of the many challenges of this field is given by the fact that the domain of each feature can be very different from one another. Common domains are: binary domains, i.e., $\mathbb{D} = \{0, 1\}$, categorical domains, e.g., $\mathbb{D} = \{\text{Red}, \text{Yellow}, \text{Blue}\}$, and numerical domains, i.e., $\mathbb{D} \subseteq \mathbb{R}$. Given a dataset $\mathcal{D}$ with $N > 1$, it is always possible to partition its instances in order to obtain a *training set* used to train a machine learning model, and a *test set* used to test a machine learning model.

**Deep Generative Modelling for Tabular Data.**

To formulate the problem of tabular data generation, the literature makes the assumptions that given a tabular dataset $\mathcal{D}$, there exists an unknown joint distribution $p_X$ over the random variables $(X_1, X_2, \ldots, X_K)$, where $X_j \in \mathbb{D}^j$, such that $\mathcal{D}$ was created by drawing $N$ i.i.d. samples. The goal of standard generative modelling is to learn from $\mathcal{D}$ the parameters $\theta$ of a generative model such that the model distribution $p_\theta$ approximates $p_X$.

While synthesising tabular data is a long-standing problem [Reiter, 2005], in this survey, we focus on methods that use deep generative models, i.e., models based on deep neural networks.

Table 1: Overview of the surveyed methods proposed for synthesising tabular data. For each work, we report (i) the model type, (ii) whether the model allows for mixed data types (marked with "✓" if so, and with "✗" otherwise), (iii) the maximum number of features used for evaluation (indicating whether the model can handle high-dimensionality data), (iv) the model's data generation time relative to the other models. In the last five columns, we indicate with "✓" (resp., "✗") the requirements that were (resp., were not) addressed, and with "•" the requirements whose resolution depends on the base DGM model upon which the Constraints-enforcing methods are built. Specifically, these requirements are: (v) utility, (vi) background knowledge alignment, (vii) statistical fidelity, (viii) privacy, and (ix) diversity. Regarding the generation time, the three indicators are assigned based on the time (in seconds) it takes to generate 10K samples, as follows: *fast* for generation time less than 0.5 s, *medium* for generation time higher than 0.5 s but less than 1 s, and *slow*, otherwise. Further, note that "—" indicates that data are not available, and "✓✓" indicates that the method provides guarantees that the domain-specific constraints are satisfied. Note that the generation time values have been collected from the surveyed papers, aggregated and had thresholds applied to them to assign them to one of the three possible indicators provided, i.e., *fast*, *medium*, and *slow*.

| Model | Model Type | Mixed Data | Max.# Features | Generation Time | Utility | Alignment | Fidelity | Privacy | Diversity |
|---|---|---|---|---|---|---|---|---|---|
| medGAN [Choi et al., 2017] | GAN & AE | ✗ | 1071 | fast | ✗ | ✗ | ✗ | ✓ | ✗ |
| TGAN [Xu & Veeramachaneni, 2018] | GAN | ✓ | 55 | medium | ✓ | ✗ | ✓ | ✗ | ✗ |
| TableGAN [Park et al., 2018] | GAN | ✓ | 32 | medium | ✗ | ✗ | ✗ | ✓ | ✗ |
| PATE-GAN [Yoon et al., 2019] | GAN | ✓ | 617 | fast | ✗ | ✗ | ✗ | ✓ | ✗ |
| ITS-GAN [Chen et al., 2019] | GAN & AE | ✓ | 45 | — | ✗ | ✓ | ✓ | ✗ | ✗ |
| CTGAN [Xu et al., 2019] | GAN | ✓ | 785 | fast | ✓ | ✗ | ✓ | ✗ | ✗ |
| TVAE [Xu et al., 2019] | VAE | ✓ | 785 | fast | ✓ | ✗ | ✓ | ✗ | ✗ |
| Neural Spline Flows [Durkan et al., 2019] | Normalising Flows | ✗ | 50 | — | ✗ | ✗ | ✓ | ✗ | ✗ |
| CWGAN [Engelmann & Lessmann, 2021] | GAN | ✓ | 36 | — | ✓ | ✗ | ✗ | ✗ | ✗ |
| OCT-GAN [Kim et al., 2021] | GAN | ✓ | 59 | medium | ✓ | ✗ | ✗ | ✗ | ✗ |
| GANBLR [Zhang et al., 2021] | GAN & BN | ✗ | 55 | — | ✓ | ✗ | ✗ | ✗ | ✗ |
| CTAB-GAN [Zhao et al., 2021] | GAN | ✓ | 55 | medium | ✓ | ✗ | ✓ | ✗ | ✗ |
| IT-GAN [Lee et al., 2021] | GAN & AE | ✓ | 59 | slow | ✓ | ✗ | ✗ | ✓ | ✗ |
| GEM [Liu et al., 2021] | Iterative NN | ✗ | 14 | fast | ✗ | ✗ | ✓ | ✓ | ✗ |
| DP-MERF [Harder et al., 2021] | Kernel-based NN | ✓ | 617 | fast | ✓ | ✗ | ✓ | ✓ | ✗ |
| GOGGLE [Liu et al., 2022] | VAE & GNN | ✓ | 168 | medium | ✓ | ✓ | ✗ | ✗ | ✓ |
| CTAB-GAN+ [Zhao et al., 2022] | GAN | ✓ | 55 | medium | ✓ | ✗ | ✓ | ✓ | ✗ |
| TabDDPM [Kotelnikov et al., 2023] | Diffusion | ✓ | 51 | slow | ✓ | ✗ | ✓ | ✗ | ✓ |
| STaSy [Kim et al., 2023] | Diffusion | ✓ | 58 | slow | ✓ | ✗ | ✗ | ✗ | ✓ |
| CoDi [Lee et al., 2023] | Diffusion | ✓ | 31 | medium | ✓ | ✗ | ✗ | ✗ | ✓ |
| GReaT [Borisov et al., 2023] | LLM | ✓ | 47 | slow | ✓ | ✓ | ✗ | ✗ | ✗ |
| TabPFGen [Ma et al., 2023] | Energy-based model | ✗ | 77 | — | ✓ | ✗ | ✗ | ✗ | ✗ |
| C³-TGAN [Han et al., 2023] | GAN & BN | ✓ | 54 | — | ✓ | ✓ | ✓ | ✗ | ✗ |
| FinDiff [Sattarov et al., 2023] | Diffusion | ✓ | 84 | slow | ✓ | ✗ | ✓ | ✓ | ✗ |
| AutoDiff [Suh et al., 2023] | Diffusion & AE | ✓ | 61 | slow | ✓ | ✗ | ✓ | ✗ | ✗ |
| DP-NTK Yang et al. [2024] | Kernel-based NN | ✓ | 617 | fast | ✓ | ✗ | ✓ | ✓ | ✗ |
| TDGGD [Jia et al., 2024] | Diffusion | ✗ | 44 | — | ✓ | ✓ | ✗ | ✓ | ✓ |
| C-DGM [Stoian et al., 2024] | Constraints-enforcing | ✓ | 109 | fast | ✓ | ✓✓ | • | • | • |
| TabSyn [Zhang et al., 2024] | Diffusion & VAE | ✓ | 48 | medium | ✓ | ✗ | ✓ | ✗ | ✓ |
| CuTS [Vero et al., 2024] | GAN | ✗ | 20 | — | ✗ | ✓ | ✗ | ✓ | ✗ |
| TabuLa [Zhao et al., 2025] | LLM | ✓ | 55 | slow | ✓ | ✗ | ✗ | ✗ | ✗ |
| DGM+DRL [Stoian & Giunchiglia, 2025] | Constraints-enforcing | ✓ | 64 | fast | ✓ | ✓✓ | • | • | • |
| TabTransGAN [Zhang et al., 2025] | Transformer & GAN | ✓ | 59 | — | ✓ | ✗ | ✗ | ✓ | ✗ |
| TabPFN [Hollmann et al., 2025] | Transformer | ✓ | 500 | slow | ✓ | ✗ | ✗ | ✗ | ✗ |
| GEM+ [Maddock et al., 2025] | Iterative NN | ✗ | 123 | fast | ✗ | ✗ | ✓ | ✓ | ✗ |

**Requirements over Synthetic Data.** While above we have the goal for standard generative modelling, we know that different use cases can lead to additional requirements, which lead to different goals for the generative task. In this survey, we review the following requirements:

1. **Utility requirement:** *Synthetic data should yield similar predictive performance to real data when used to train machine learning (ML) models for the same task, such as classification or regression.* For instance, given a healthcare dataset like WiDS [Matthys et al., 2021], which contains medical records of patients from the first 24 hours of intensive care, and the target column specifies whether a patient has been diagnosed with diabetes mellitus, training a classifier on synthetic data should yield comparable (or better) performance on the real test set to that obtained by training the classifier on real data of the same size. Hence, given an ML model $m$ and a predictive performance metric $r$ (e.g., accuracy), maximising utility entails learning from $\mathcal{D}$ the parameters $\theta$ of a generative model such that training $m$ on the dataset $\mathcal{D}'$ sampled from $p_\theta$ leads to the best score according to $r$.

In addition to predictive performance, utility can be broadly defined as the preservation of valid statistical inferences [Raghunathan et al., 2003]. For instance, in differentially private settings, Rosenblatt et al. [2023] quantify this as the likelihood that the empirical conclusions derived from the real data would not change if synthetic data were used instead. In these cases, maximising this statistical utility boils down to maximising this likelihood.

2. **Alignment requirement:** *Synthetic data should align[1] with any known (user-provided) domain-specific knowledge.* The knowledge can be expressed in multiple ways: simply as constraints on the range of values that a single feature can assume, or as more complex constraints capturing relations between the features. Continuing the example above, if the *minimum* and the *maximum haemoglobin level* are two of the features recorded for each patient, then clearly the real data do not contain any records for which the value of the *minimum* is higher than the value of the *maximum level*. This type of constraint is easily captured using a linear inequality. Hence, given a set of constraints expressing some background knowledge about the sample space of $p_X$, i.e., stating which samples are admissible and which are not, the goal is to learn the parameters $\theta$ of a generative model such that (i) the model distribution $p_\theta$ approximates $p_X$, and (ii) the sample space of $p_\theta$ is compliant with the constraints.

3. **Fidelity requirement.** *Synthetic data should preserve the statistical properties of the real data.* Continuing with our example, if we are maximising fidelity on the WiDS dataset, then the distribution of the synthetic values of the *minimum haemoglobin level* feature should resemble the real distribution of the same feature. Hence, in this case, the goal is to learn parameters $\theta$ such that the marginals of $p_\theta$ are as similar as possible to the corresponding marginals of $p_X$, and the joint distribution $p_\theta$ should closely follow $p_X$.

4. **Privacy requirement.** *Synthetic data should present minimal risk of disclosing sensitive attributes and of re-identifying individuals from the real data.* Continuing the earlier example, the WiDS dataset contains sensitive features such as the patient's *age*, whether the patient had an *elective surgery*, and whether the patient received a *leukemia* diagnosis. If a rare combination of values for these features is retained in the synthetic data, then a patient might be identified, hence failing to meet privacy requirements. Hence, in this case, the goal is to learn the parameters $\theta$ such that it is impossible to sample a data point from $p_\theta$ that would allow for the re-identification (or for disclosing sensitive attributes) of a real data point.

5. **Diversity requirement.** *Synthetic data should cover the full variability of the real data distribution.* While closely related to fidelity, diversity focuses on a different aspect of the data distribution. Specifically, diversity ensures that the model does not omit regions of the output sample space, which is what would happen in the case of a mode collapse, for example. Continuing the earlier example, it is likely that the WiDS dataset contains under-represented patient groups, such as young adults with specific co-morbidities or rare physiological responses, e.g., captured as extreme but valid laboratory test result values. In such cases, failing to meet the diversity requirement might only produce "average" patient profiles. Thus, the goal is to learn parameters $\theta$ such that the support of $p_\theta$ fully covers the support of $p_X$, which would allow for rare feature values or combinations of values to be synthesised (with appropriate probability).

---

[1]To further clarify our categorisation, we acknowledge the possible terminology overlap with previous papers. For example, Xu et al. [2019] mention that the discriminator of CTGAN is trained such that the synthetic data *aligns* with the real data. However, in this case, the term *align* is used informally and refers to the process of matching the real data distribution, which falls under the fidelity requirement in our survey. The formal definition of alignment has been introduced in [Stoian et al., 2024], and is reproduced here to convey alignment with the given background knowledge.

Table 2: Overview of the evaluation metrics typically used for the five synthetic tabular data requirements. The formula and terminology definitions for each metric are provided in Appendix A.

| Requirement | Evaluation Metric | |
|---|---|---|
| Utility | Accuracy | |
| | F1-score | |
| | Root mean square error (RMSE) | |
| | Explained variance | |
| | Epistemic parity | |
| Alignment | Constraint violation rate (CVR) | |
| | Constraint violation coverage (CVC) | |
| | Sample-wise constraint violation coverage (sCVC) | |
| Fidelity | Feature-wise | Wasserstein distance |
| | | Kolmogorov-Smirnov test statistic |
| | | Jensen-Shannon divergence |
| | | Total variation distance |
| | Pair-wise | RMSE differences in mutual information |
| | | MAE differences in mutual information |
| | | Correlation difference |
| | Joint | Likelihood fitness score |
| | | $\alpha$-Precision |
| | | Density |
| | | Clipped density |
| Privacy | AUCROC (for membership attacks) | |
| | Precision (for attribute disclosure attacks) | |
| | Recall/Sensitivity (for attribute disclosure attacks) | |
| | Distance to the closest record (DCR) | |
| | Nearest neighbour distance ratio (NNDR) | |
| | k-anonymity | |
| Diversity | $\beta$-Recall | |
| | Coverage | |
| | Clipped coverage | |

## 3 Requirements and Evaluation Metrics

To systematically evaluate the quality of synthetic tabular data and guide the selection of appropriate synthesisers, in this section, we detail each of the five requirements, along with the specific evaluation protocols and metrics used to quantify them, and discuss their significance in real-world applications.

For each of the five requirements above, Table 2 lists the metrics that are commonly used to evaluate the generated data. Further, in Appendix A, we provide the formula for each of these metrics, along with the terminology definitions.

### 3.1 Utility

Determining whether synthetic data can replace real training data in downstream tasks has been the most common evaluation approach (e.g., [Choi et al., 2017; Park et al., 2018; Kotelnikov et al., 2023; Borisov et al., 2023]). The main reason for this is that synthetic data are often generated to augment or create datasets for real-world applications, including high-stakes fields like finance and healthcare. In these scenarios, generating synthetic samples for rare events, e.g., fraudulent transactions and specific medical diagnoses [Matthys et al., 2021], can help with class balancing and ultimately improve the quality of the predictions in downstream tasks. Therefore, utility still remains a key requirement for synthetic data and benchmark for comparing synthesisers.

**Evaluation Protocol and Metrics.** The most common protocol used to evaluate the utility of generative models is the Train on Synthetic, Test on Real (TSTR) Protocol (see, e.g., [Esteban et al., 2017]). The procedure is to train a suite of ML models (e.g., a support vector machine, XGBoost) on the synthetic data, and then test their performance on a real test set, reporting the performance using different metrics on the ground of the task defined by the dataset of interest: for classification (resp., regression) datasets, standard metrics include accuracy and F1 score (resp., root mean square error and explained variance).

Beyond predictive performance (TSTR), foundational work on classical statistical, non-deep learning methods for generating fully synthetic data in [Raghunathan et al., 2003] aimed to preserve the users' ability to obtain valid statistical inferences without compromising the confidentiality of survey respondents. Further, in the specific case of differentially private settings, recent work by Rosenblatt et al. [2023] similarly argues that synthetic data must also support valid inference, ensuring that statistical conclusions derived from real data are preserved. To measure this, they introduce a metric called epistemic parity, which evaluates whether empirical conclusions of peer-reviewed papers on real, publicly available data can be reproduced on synthetic datasets. To this end, in the context of synthesising data intended for scientific discovery, Montoya Perez et al. [2024] emphasise that evaluating the validity and power of statistical tests is essential.

## 3.2 Alignment

Brought to the attention of the research community more recently [Chen et al., 2019] than the other requirements, background knowledge alignment is an important condition for synthetic data when evaluating its realism. This requirement is important in practice, as many real-world applications have domain-specific knowledge that the synthetic data must satisfy. For example, synthetic patient data in healthcare must adhere to physiological constraints, ensuring a recorded minimum value for an indicator (like blood pressure or haemoglobin level) is not higher than the maximum value recorded. Similarly, in finance, it is important for generated market data to comply with known economic constraints. In resource management, synthetic inventory and demand data must not violate logistical constraints, such as production capacity limits or non-negative stock levels.

**Evaluation Protocol and Metrics.** In [Stoian et al., 2024], three metrics have been proposed to evaluate alignment: (i) the Constraint Violation Rate (CVR), which computes the percentage of samples that do not satisfy at least one of the constraints in the available set of constraints, (ii) Constraint Violation Coverage (CVC), computing the percentage of constraints that have been violated at least once by the sample set, and (iii) the sample-wise Constraint Violation Coverage (sCVC) which determines the average percentage of samples violating each of the constraints. Out of these three metrics, CVR has also been used in [Jia et al., 2024], albeit called feasibility rate there.

## 3.3 Fidelity

While statistical fidelity is not a good predictor of downstream task performance [Hansen et al., 2023], it can provide useful observations on how the synthetic data compares to the real data. Fidelity is indeed one of the requirements often evaluated in early works on synthesising tabular data and it is crucial for applications such as economic modelling and census data release. In these areas, preserving the statistical properties of the real data distribution, like demographics, allows for accurate downstream sociological and economic analyses. Similarly, generating data that captures the statistical complexity of real-world data is valuable in other domains, such as the robust testing of data processing software.

**Evaluation Protocol and Metrics.** Being the longest-standing requirements, a large number of methods were developed to evaluate the statistical fidelity of synthetic data w.r.t. real data.[2] These can be categorised into (i) feature-wise, (ii) pair-wise, and (iii) joint evaluations.

1. **Feature-wise evaluation**: compares synthetic and real data feature by feature, using different metrics based on feature type. For continuous features, Wasserstein distance and the Kolmogorov-

---

[2]In this survey, we focus only on quantitative evaluations.

Smirnov test statistic are common, while Jensen-Shannon divergence or total variation distance are common metrics for discrete features. The difference in treatment is supported empirically in [Zhao et al., 2021], which found that Jensen-Shannon divergence is unstable for measuring the statistical similarity between continuous features' distributions, especially with no overlap between synthetic and real data.

2. **Pair-wise evaluation**: investigates how well relationships between pairs of features are preserved in the synthetic w.r.t. the real data. Common metrics include: (i) RMSE (and MAE) differences in mutual information between pairs of features, and (ii) correlation difference, using Pearson's coefficient for continuous pairs, Theil's uncertainty coefficient for discrete pairs, and the correlation ratio between discrete-continuous pairs of features.

3. **Joint evaluation**: compares the joint distribution of the synthetic samples against the real samples' distribution. It is more difficult to determine how similar are two (potentially high-dimensional) joint distributions. Thus, such an analysis is often conducted in a controlled environment, e.g., on simulated data, rather than real-world data. Relevant evaluation methods include measuring: (i) the likelihood fitness score, separately for real and synthetic data (as in [Xu et al., 2019]), (ii) $\alpha$-precision [Alaa et al., 2022], which is a variant of precision representing the fraction of synthetic samples that resemble the most typical fraction $\alpha$ of real samples, (iii) density (defined in [Naeem et al., 2020]), which improves upon the standard precision metric by counting the number of real manifolds each synthetic sample falls within, and (iv) clipped density, which has been proposed by Salvy et al. [2025] as an improvement on the density metric in terms of robustness, clipping the radius of the nearest-neighbour manifolds used to measure the density to the median of the distances to the $k$-th nearest neighbour.

### 3.4 Privacy

Synthetic data are particularly valuable in privacy-sensitive domains. Thus, there has been growing focus on generating data without revealing sensitive information that could identify entities from the original dataset. For example, sharing synthetic patient records for medical research allows for collaboration without risking the re-identification of individuals or the disclosure of sensitive health information. Similarly, privacy is an important requirement in education analytics, where synthetic educational data allows for studying student performance trends without exposing individual student identities or sensitive background information. Finance is another key domain where privacy is a concern and synthesising transaction or credit data to be released for academic or commercial research must be done in a way that protects customer confidentiality.

**Evaluation Protocol and Metrics.** Privacy is typically evaluated by assessing the risk of attacks succeeding in targeting sensitive information. The most common attacks are: (i) **membership inference attacks**, which use ML classifiers or black-box attacks (e.g., see [Chen et al., 2020]) to determine whether an entity was part of the set used to train a model, (ii) **attribute disclosure attacks**, which try to gain access to sensitive attributes of an entity from the real dataset, and (iii) **re-identification attacks**, which try to map a synthetic data point back to the original dataset. To measure the risk of the attacks being successful, for (i), AUCROC is often reported for the trained attack models (typically one for each class of a feature of interest). However, this metric can be misleading, as it might suggest that a model is secure even if the privacy of a few users has been confidently breached or, conversely, imply that an attack is successful based on unreliable scores. To this end, Carlini et al. [2022] argue that membership inference attacks should be evaluated using the true-positive rate (TPR) at low false-positive rates (FPR), which is capable of detecting such high-confidence attacks that aggregated metrics, such as AUCROC, might overlook. For (ii), precision and sensitivity (i.e., recall) are measured while varying the number of attributes known to the attacker, and, for (iii), common metrics include: the distance to the closest record (DCR) [Zhao et al., 2021] (computed as the minimum L2 distance from a synthetic data point to each real data point), the nearest neighbour distance ratio (computed as the ratio between the distances to the closest and second-closest real neighbours for a synthetic record), but also k-anonymisation, delta presence, and the identifiability score, as seen in [Jia et al., 2024].

Finally, methods using differential privacy [Dwork, 2006] introduce noise during training to reduce retention of original data features. Privacy is then evaluated using different privacy budget values, which determine the amount of added noise and balance utility and privacy (e.g., see [Park et al., 2018]).

### 3.5 Diversity

While fidelity ensures that generated samples resemble the real ones, diversity ensures that the synthetic data cover the full variability of the real data distribution [Naeem et al., 2020]. This plays an important role in avoiding mode collapse, where a synthesiser produces samples that fit well within the data distribution but which are repetitive, failing to represent rare feature combinations or outliers which are present in the real data. This is of particular importance in healthcare as described earlier, but also in domains such as finance, e.g., for detecting fraud patterns.

**Evaluation Protocol and Metrics.** Typically, diversity is evaluated by determining the proportion of the real data manifold that is represented by the generated data. There are multiple ways to compute this, with the most common metrics being: (i) $\beta$-**recall** [Alaa et al., 2022], which is a variant of recall evaluating whether the synthetic data covers the variability of the real data (e.g., as used in [Liu et al., 2022]), (ii) **coverage** (defined in [Naeem et al., 2020] and used in [Kim et al., 2023; Kotelnikov et al., 2023]), which improves upon the standard recall metric by building nearest neighbour manifolds around the real samples, instead of around the synthetic samples, to ultimately measure the proportion of real samples whose neighbourhoods contain at least one synthetic sample, and (iii) **clipped coverage**, which has been proposed by Salvy et al. [2025] as an improvement on the coverage metric in terms of robustness (e.g., unlike the standard metric which only checks for the presence of synthetic samples within the real manifolds, clipped coverage counts them and bounds their contributions to effectively reflect the real data distribution rather than just its support).

### 3.6 Relationships among the Requirements

As emphasised by Naeem et al. [2020], assessing a generative model is an inherently difficult task, as no single metric can capture all the desired properties of synthetic data. And, while the requirements described in Sections 3.1-3.5 are presented individually, they are not orthogonal. In fact, these requirements' relationships vary from mutual reinforcement to mutual tension, possibly leading to tradeoffs. Thus, forming an understanding of these relationships is crucial for practitioners to select the appropriate synthesiser for their specific use case.

**Fidelity and Utility.** Intuitively, higher statistical fidelity, where the distribution of the synthetic data $p_\theta$ closely approximates the distribution of the real data $p_X$, often yields higher utility. However, this is not always the case, as a high statistical fidelity does not guarantee optimal utility in downstream tasks and vice-versa. Indeed, as shown by Hansen et al. [2023], fidelity alone is insufficient for assessing the synthetic data's utility, and should not be used as an indicator for it.

**Alignment and Fidelity.** The relationship between these two requirements can be seen as hierarchical. A model with "perfect" fidelity would implicitly satisfy all background knowledge constraints present in the real data, assuming such data do not contain any errors. However, high alignment does not necessarily imply high fidelity. In spite of this, alignment plays a vital role for any model that is to be deployed in domains where constraints must be guaranteed [Stoian et al., 2024], whereas fidelity often remains an approximation.

Furthermore, it is still possible to have real data that contains erroneous values (e.g., swapped minimum and maximum values due to human error). In such cases, the constraints can be used to validate and correct the data before synthesis. Indeed, if these errors are not addressed, a model that strictly satisfies the alignment requirement may inevitably deviate from statistical fidelity measured in this case against the corrupted real data.

**The Privacy Trade-off (Privacy, Fidelity, and Utility).** The tension between privacy and utility is perhaps the most critical trade-off in tabular data synthesising. In particular, any method that claims to

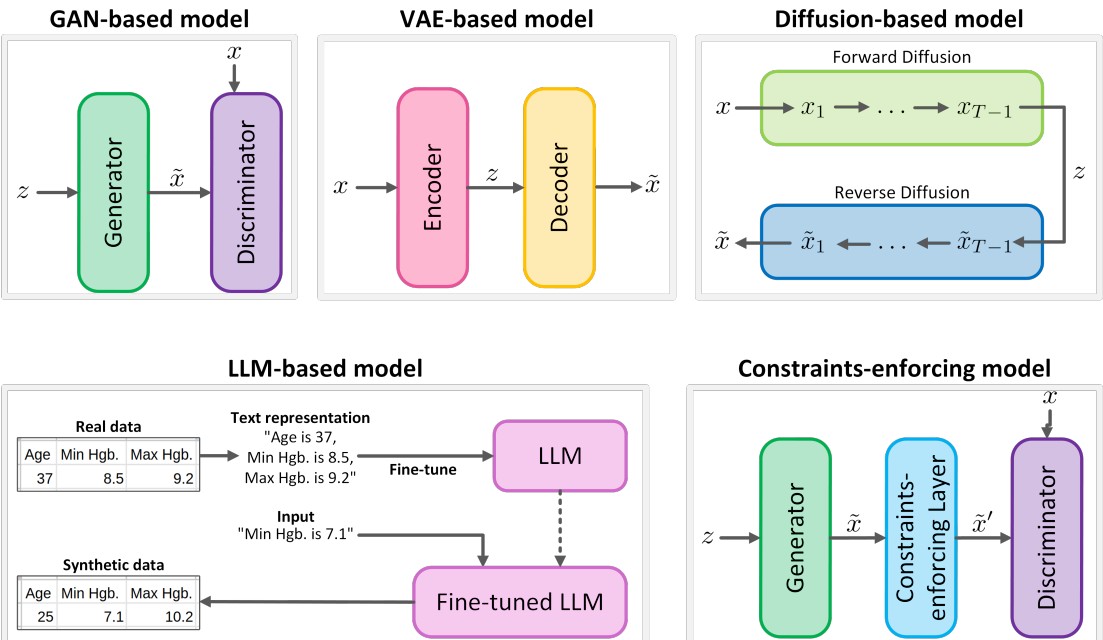

Figure 2: Visualisation of common model types used for synthesising tabular data. Here, $x$, $z$, and $\tilde{x}$ denote a real sample, noise sample, and synthetic sample, respectively; $\tilde{x}'$ is a sample modified by layer-based models; $x_i$ is the sample at step $i$ of forward diffusion (for $i$ in $\{1, \ldots, T\}$, where $T$ is the maximum number of diffusion steps); and $\tilde{x}_i$ is the sample at step $i$ of reverse diffusion.

be preserving privacy must balance privacy and utility jointly rather than separately Montoya Perez et al. [2024]. To satisfy differential privacy, for example, noise must be added during sample generation. And, as more such noise is added (i.e., to better protect the data subjects), statistical fidelity decreases, which often leads to lower downstream performance (i.e., utility). It is thus important for synthesising methods to determine the acceptable extent of utility to be sacrificed in order to achieve robust privacy guarantees.

**Diversity and Fidelity.** Intuitively, fidelity and diversity are associated with precision and recall, respectively [Sajjadi et al., 2018; Naeem et al., 2020]. Models suffering from mode collapse may still produce high-fidelity samples (i.e., high-precision), but fail to cover the full variability of the real data distribution (i.e., low diversity or low recall). Conversely, maximising diversity may result in generating outliers or noisy samples, which often leads to lower fidelity. Thus, these two metrics are seen as complementary trade-offs, as maximising one without the other leads to suboptimal synthetic data. Further, in an effort to disentangling fidelity from diversity, different metrics have been proposed to measure each of the two requirements, which were often improved precision and recall variants as first presented in [Kynkäänniemi et al., 2019]), and subsequently in Naeem et al. [2020]; Alaa et al. [2022]). Nevertheless, the recent position paper by Räisä et al. [2025] argues that current metrics for measuring fidelity and diversity are flawed and advocates for metrics with meaningful absolute values, rather than just relative comparisons, to truly determine if a model is useful in practice.

## 4 Methods

Having established the key requirements for synthetic tabular data, we now survey the deep generative models designed to meet them. To provide a clear structural overview, we categorise these approaches based on their underlying generative paradigm: generative adversarial networks (GANs), diffusion models, transformers and large language models (LLMs), variational autoencoders (VAEs), constraints-enforcing methods, along with other deep learning methods (e.g., approaches stemming from other domains but which

have been adapted to tabular data synthesis). Finally, we also discuss briefly non-deep learning alternatives.

## 4.1 Generative Adversarial Network-based Methods

### 4.1.1 GAN-based models

Many of the works surveyed here are based on the original Generative Adversarial Network (GAN) [Goodfellow et al., 2014], which relies on two neural networks: (i) a *generator*, which given a noise vector returns a synthetic datapoint and hence learns the real data distribution, and (ii) a *discriminator*, which given a data point (either real or synthetic) classifies it as either real or synthetic and hence learns to distinguish between synthetic and real data. The GAN jointly trains these two models in an adversarial framework, where the generator's objective is to produce samples that confuse the discriminator. See the first schema from the left in Figure 2 for a simple abstraction of GAN-based models.

CWGAN: Building on the seminal Wasserstein GAN (WGAN) [Arjovsky et al., 2017], which used the Wasserstein distance as a loss function for a GAN model, Engelmann & Lessmann [2022] proposed CWGAN as an oversampling framework for class balancing. CWGAN generates synthetic samples for underrepresented classes, augmenting the training set. Results indicate that oversampling is beneficial primarily for strongly non-linear datasets. OCT-GAN: proposed by Kim et al. [2021], incorporates neural ordinary differential equations (neural ODEs) in their GAN through the introduction of a new layer that allows for the extraction of a sequence of hidden vector representations given an input sample, i.e., the hidden evolution trajectory of the sample. The trajectory is used to help the discriminator decide whether a sample is real or synthetic. While improving utility, the usage of ODEs makes OCT-GAN slower than most GAN-based models. TGAN [Xu & Veeramachaneni, 2018] is an early GAN-based model that proposes using long short-term memory networks with an attention mechanism to synthesise data sequentially, feature by feature, in order to model tabular data of mixed types. Comparing with conventional statistical models and showing better fidelity performance, TGAN contributed to the popularity of the GAN-based approaches for tabular data.

CTGAN: focused on better modelling the types of the variables in an effort to increase fidelity. Indeed, the authors propose a new preprocessing method, which is column-type-specific and models discrete features in the continuous space via Gumbel-Softmax transformations, while it employs mode-specific normalisation via a variational Gaussian mixture model applied for each continuous features. Thanks to these transformations, CTGAN is able to avoid mode collapse, which often burdens other GAN-based models, like medGAN [Choi et al., 2017] and TableGAN [Park et al., 2018]. CTAB-GAN: Building upon CTGAN, CTAB-GAN [Zhao et al., 2021] enhances statistical fidelity by addressing data imbalance and long-tail issues. It introduces a conditional vector to handle mixed-type features and multi-modal continuous variables, extending support beyond discrete and continuous features. Further, it supports mixed-type features: which are features that might have highly-recurring discrete values, but also continuous values across the data points (e.g., a feature *Credit Card Transaction USD* might often have value 0, but it can also exceed 500). The benefit of providing support for mixed-type features is reflected in the fidelity performance, with CTAB-GAN outperforming CTGAN. CTAB-GAN+ [Zhao et al., 2022] is an extension of CTAB-GAN, which addresses privacy concerns by training a discriminator using differential private-SGD [Abadi et al., 2016]. Importantly, CTAB-GAN+ demonstrates high fidelity performance.

TableGAN: Another model that marked an important milestone for deep generative modelling of tabular data is TableGAN [Park et al., 2018], which, unlike medGAN, can model continuous features. While it is based on a standard GAN model, consisting of a generator and a discriminator, TableGAN has an additional component: a classifier that predicts the value of the target feature for synthetic samples using semantic integrity constraints learned from the real data. As exemplified in the original paper, a sample with values of 60 and 1 for features *Cholesterol* and *Diabetes*, respectively, does not align with the background knowledge, as the cholesterol level is too low for the target feature to indicate a diabetes diagnosis. Focusing on ensuring privacy, the authors argue for the use of synthesiser models, which do not learn bijective relationships between real and fake data, making them less vulnerable to re-identification attacks, unlike prior perturbation or anonymisation methods. Indeed, TableGAN shows that it can synthesise tabular data with a low risk of re-

identification attacks, membership inference attacks and attribute disclosure attacks to succeed. PATE-GAN: Another model that is based on a GAN architecture, but alters it for preserving privacy, is PATE-GAN [Yoon et al., 2019]. Rather than using the standard GAN discriminator, PATE-GAN relies on a modified Private Aggregation of Teacher Ensemble (PATE) [Papernot et al., 2017] mechanism to ensure that the discriminator is differentially private. More specifically, a number of teacher-discriminators are created and trained to improve their loss w.r.t. the generator, which adjusts its loss according to the single student-discriminator in the mechanism. In turn, the student-discriminator adjusts its loss according to the teacher-discriminators and allows for backpropagation to the generator, closing the loop. Critically to the privacy requirement, specialising each of the teacher-discriminators on small partitions of the training set ensures that the effect of individual samples is small and, thus, higher differential privacy.

CuTS: Vero et al. [2024] proposed CuTS as a customisable framework, which can adapt to user-defined specifications, such as background knowledge constraints, but also to differentially private training. In the private setting, CuTS builds on the DP iterative framework from [McKenna et al., 2022], but uses its own GAN-based generator, which is equipped with a mechanism that matches the generator's output distribution to the real data distribution by comparing marginals. Additionally, CuTS can incorporate background knowledge in the form of propositional logical constraints that each data point must satisfy via a new loss term. Further, rejection sampling is used to ensure that the model's outputs satisfy the constraints. Although the framework works only with discrete features, it takes a step towards synthesisers for tabular data that account for multiple requirements, including alignment and privacy.

### 4.1.2 GAN and BN-based models

GANBLR: Zhang et al. (2021) construct the generator and discriminator as Bayesian networks (BNs), which are able to encode known feature interactions. The background knowledge inclusion leads to better utility than standard GANs. $C^3$-TGAN: Similarly to GANBLR, $C^3$-TGAN [Han et al., 2023] uses Bayesian networks to capture background knowledge. However, while GANBLR models only discrete features, $C^3$-TGAN can handle both discrete and continuous features by following the CTGAN [Xu et al., 2019] approach. To incorporate explicit attribute correlations and property constraints from background knowledge, it represents constraints as control vectors (similar to the conditional vectors in CTGAN) and uses them to guide training, ensuring better alignment.

### 4.1.3 GAN and AE-based models

ITS-GAN: proposed for tabular data augmentation ITS-GAN [Chen et al., 2019] is an early method advocating for the use of background knowledge. The knowledge it can express is of two types: (i) rules stating that the value of one feature uniquely determines the value of another (e.g., the feature *Position* determines the feature *Salary*), and (ii) rules stating that a specific value for a set of features determines the specific value for another (e.g., if *Position = "CEO"* then *Salary = "2M"*). To encode the first types of rules, they train on the real data an autoencoder (AE) for each rule to output the value of the dependent features given the values of the independent ones. Then, at training time, the GAN generator is trained to minimise the difference between its output and one of the AEs, while also being penalised if it violates any rule of the second type. Additionally, the discriminator receives as input the difference between the AE's outputs and the synthetic samples, thus being able to use this information to assess the sample.

medGAN: Proposed by Choi et al. [2017] for synthesising healthcare datasets, medGAN is one of the pioneering models for synthesising tabular data using deep generative models. medGAN generates distributed representations of health records using the generator of a GAN-based model, then decodes these representations using an autoencoder pretrained on the real data, and passes the decoded representations to the discriminator component of the GAN, which is trained to distinguish the synthetic from the real data. In terms of privacy, medGAN showed low attribute disclosure risks, except when an attacker focused on a small number of data points. IT-GAN: Similar to OCT-GAN but designed to enhance privacy robustness, IT-GAN [Lee et al., 2021] employs a generator based on neural ODEs to balance utility and privacy protection. Instead of directly synthesising samples in the input space, the generator produces hidden representations, as seen also in medGAN. Notably, it ensures that the hidden representation space matches the input dimensionality, enabling the application of ODEs.

### 4.1.4 GAN- and Transformer-based models

TabTransGAN [Zhang et al., 2025] achieves a performance comparable to GReaT [Borisov et al., 2023] by combining the strengths of Transformer [Vaswani et al., 2017] and GAN-based models, which allows it to model both contextual and structural relationships across all features. Specifically, TabTransGAN adopts a GAN architecture in which the generator is based on CTGAN, while the discriminator replaces the standard design with a Transformer architecture. This enables it to better capture both feature-wise distributions and dependencies between the features, leading to more accurate discrimination between synthetic and real data.

## 4.2 Diffusion-based Methods

### 4.2.1 Diffusion-based models

TabDDPM: Motivated by the their success in computer vision, Kotelnikov et al. 2023 introduced TabDDPM: a diffusion-based model for synthesising tabular data. Popular for its high utility results, TabDDPM is able to model both discrete and continuous features, but separately, by using multinomial [Hoogeboom et al., 2021] and Gaussian diffusion models [Sohl-Dickstein et al., 2015], respectively. For a simple abstraction on how a diffusion model can be used to generate data, see the central schema in Figure 2. STaSy: Proposed by Kim et al. [2023], STaSy is based on a score-based generative model (SGM), which was shown to be a competitive alternative to diffusion-based models in [Ho et al., 2020]. Contrarily to TabDDPM, StaSy estimates the score function (gradient of log-probability) instead of explicitly modelling the forward and reverse Markov processes. As the loss is difficult to train, STaSy uses a self-paced learning strategy, which starts with a subset of training data—yielding a stable, low loss—and gradually expands to the full dataset. Evaluated on 15 real-world datasets against 7 baselines (mostly GAN-based models), STaSy achieves high utility. Due to the forward and backward passes, both StaSy and TabDDPM suffer from slow sampling times. Moreover, the base SGM model is known to be unstable in high-dimensional settings. CoDi: Motivated by the difficulty of modelling discrete features, Lee et al [2023] proposed CoDi, a framework comprising co-evolving diffusion models that separately handle continuous and discrete features as in TabDDPM [Kotelnikov et al., 2023], with the difference that the two models condition on each other during training. At each training step, the discrete (resp., continuous) diffusion model takes the perturbed sample from the continuous (resp., discrete) one as input, and both models are conditioned on both samples from the previous step during the denoising process. Slower than most GAN-based models (excluding OCT-GAN), CoDi is faster than STaSy and TabDDPM, while also obtaining higher utility on mixed types datasets.

TDGGD: The first framework encoding background knowledge about feature relations into diffusion-based models is TDGGD [Jia et al., 2024]. The relations can express lower and upper bounds over single features or their sum. Rather than explicitly modelling the relations, TDGGD models whether the features are part of such relations or not, which only gives an indication of hidden relations between the columns to the model. FinDiff: Motivated by the need to preserve the statistical properties of real data of mixed types in real-world financial contexts, Sattarov et al. [2023] proposes FinDiff, which is capable of generating mixed-type tabular financial data using a diffusion model to capture high-dimensional dependencies. Unlike another popular diffusion model, TabDDPM [Kotelnikov et al., 2023], which uses one-hot encoding for categorical features, FinDiff employs an embedding encoding for better handling of mixed data types. This leads to higher fidelity performance compared to the baselines, indicating that FinDiff effectively captures both feature-wise and joint distributions of the data.

### 4.2.2 Diffusion and AE-based models

AutoDiff: Proposed in [Suh et al., 2023], AutoDiff is a diffusion-based model designed to preserve statistical fidelity by modelling the joint distribution via an autoencoder, rather than modelling the distributions of individual features separately, as seen in GAN- and diffusion-based models like CTGAN and TabDDPM. More precisely, AutoDiff (i) uses the autoencoder to learn the distribution of the features (which can be of mixed types) in a continuous space, and then (ii) passes these continuous representations to the diffusion model, which generates latent representations that are then decoded back into representations in the original feature space. Moreover, like CTAB-GAN, it introduces an approach to handle mixed-type features by

adding a new feature to the autoencoder that encodes the frequency of recurring values in a mixed-type feature.

### 4.2.3 Diffusion and VAE-based models

TabSyn: Introduced by Zhang et al. [2024], TabSyn trains a score-based diffusion model, like STaSy, but in a joint VAE-learned space of numerical and categorical features. Notably, TabSyn significantly reduces the runtime of diffusion-based models while outperforming baselines w.r.t. fidelity and utility.

## 4.3 Transformer and Large Language Model-based Methods

GReaT [Borisov et al., 2023] represents an answer to the abundance of works that convert tabular data into numerical representations thus losing the contextual connections between the features, which very often carry semantic meaning. Indeed, GReaT performs the following steps (also illustrated in Figure 2): first, the tabular data undergoes a textual encoding process, converting it into text. Next, a feature order permutation step is applied. The resulting sentences are then used to fine-tune the large language models (LLMs) [Vaswani et al., 2017]. At sampling time, either a single feature name or arbitrary feature-value pairs are given as input to the LLM, which then completes them (note that this allows for arbitrary conditioning). Using this approach, GReaT outperforms most of the baselines w.r.t. utility, but its data sampling time is notably high. To address this problem, TabuLa [Zhao et al., 2025] introduces a method to reduce the length of the generated token sequences. Unlike GReaT, which relies on large language models pre-trained on natural language processing tasks, TabuLa starts with a randomly initialised model and iteratively fine-tunes it on tabular data synthesis tasks. Another key distinction is that TabuLa does not use GReaT's feature permutation mechanism. Instead, it targets scenarios where arbitrary feature conditioning is not needed and assumes that feature values appear in a consistent order across all data points. To this end, the authors tokenise the dataset feature-wise, ensuring that each generated token can be mapped to a specific feature based on its absolute position in the row. Specifically, for each feature, they determine the longest token sequence across the dataset and pad all sequences (corresponding to that feature) to this length during training, Using this approach, TabuLa manages to significantly reduce the training time compared to GReaT. However, its sample generation time is still considerably higher than GAN-based models such as CTGAN or TableGAN.

### 4.3.1 Tabular Foundational Models

TabPFN Bringing a shift in tabular data synthesis, away from training synthesisers on each new dataset, the tabular prior-data fitted network (TabPFN) Hollmann et al. [2025] leverages in-context learning [Brown et al., 2020] mechanism of transformer-based methods and is pre-trained on a large collection of synthetic datasets rather on than large collections of publicly available datasets. In this way, TabPFN avoids issues such as privacy breaches, copyright infringements, and cross-contaminating the training data with test data. To generate synthetic datasets, it uses structural causal models, which are able to represent causal relationships in the data via a directed acyclic graph. Although primarily a predictive model building on its preliminary version from [Hollmann et al., 2023], it is important to note that TabPFN can also generate tabular data by approximating the joint distribution of features and targets, and is able to outperform the classical methods, such as tree-based baselines, on small- or even medium-sized datasets of up to 10K samples. Given its transformer-based architecture, sampling time during inference is high, but the model intrinsically does not require training, as it can adapt and do inference on unseen real-world datasets.

## 4.4 Variational Autoencoder-based Methods

### 4.4.1 VAE-based models

TVAE [Xu et al., 2019], proposed along with CTGAN, was designed to check whether CTGAN's generator's lack of access to the real data during training makes it weaker compared to models that do use these data for training their generator, such as models based on Variational Autoencoders (VAEs) [Kingma & Welling, 2014]. Indeed, TVAE's architecture showed an improvement over CTGAN in terms of fidelity. However, like

CTAB-GAN+, the authors also considered the impact of their proposed models w.r.t. privacy and noted that TVAE's access to the training data might make it unsuitable in downstream tasks where sensitive data are present.

### 4.4.2 VAE and GNN-based models

GOGGLE: [Liu et al., 2022] is a different approach which integrates knowledge about pairwise feature dependencies by encoding them in a graph where each feature is a node and an edge exists if a relation between the two features is known. New relations can be learnt using a message passing mechanism from graph neural networks (GNNs), which is jointly trained with a VAE-based architecture, where only relationships prioritised by the learned graph influence feature generation. Due to its relational learning graph-based component, it might be difficult to scale GOGGLE to datasets of higher dimensionality.

### 4.5 Constraints-enforcing Methods

C-DGM: The first method to constrain deep generative models and guarantee the constraints satisfaction was proposed in [Stoian et al., 2024]. The constraints can capture any set of linear inequalities over the feature space. Differently from the methods surveyed above, this approach relies on a layer to be added right before the sample output layer, which restricts the output space of the model to coincide with the space defined by the constraints. The layer can be added on top of any deep generative model (DGM), and the resulting models are called C-DGMs. For a visual representation on how the layer-based methods can be added on a GAN model, see the right-most schema in Figure 2. As the layer is differentiable and acyclic, it can be added both at inference and training time, while also improving the utility of the model. DGM+DRL: The above layer was further extended in [Stoian & Giunchiglia, 2025], in order to capture constraints as expressive as disjunctions over linear inequalities (which allow for expressing relations like "if the sum of two features is greater than 10 then the difference of other two features should be lower than 5"), thus modelling non-convex and even disconnected output spaces. The new layer is called Disjunctive Refinement Layer (DRL), and a DGM constructed using DRL is called DGM+DRL. Just as C-DGM, DGM+DRL is able to guarantee alignment with the background knowledge, while also improving utility across all considered baselines.

### 4.6 Other Deep Learning Methods

**Normalising flows-based models.** Neural Spline Flows: Rather than using the common transformations found in normalising flows, Durkan et al. [2019] proposed using a fully-differentiable module based on monotonic rational-quadratic piecewise functions. This adaptation, combined with the inherent properties of normalising flows, which model the data as the output of an invertible differentiable transformation of a noisy sample drawn from a known distribution, can help synthesise high-fidelity tabular data. The authors note that such a result is conditional on having enough training data compared to the datasets dimensionality.

**Energy-based models.** TabPFGen [Ma et al., 2023] is a generative model that uses a pretrained network as an energy-based model for data augmentation and class balancing. Although its capabilities are limited to low-dimensional inputs (it was tested on datasets with maximum 10 features), TabPFGen demonstrates good utility performance in the data augmentation task by simply using a pretrained model compared to specialised GAN-, VAE-, and diffusion-based models.

**Iterative privacy-preserving NN models.** GEM: Addressing the computational bottleneck of differentially private query release methods such as MWEM [Hardt et al., 2012], Liu et al. [2021] propose new generative networks with the exponential mechanism (GEM), which can be incrementally updated without retraining from scratch. The novelty of this approach is its capability to represent the data distribution using a set of generative models parametrised by neural networks. Unlike the prior methods, this approach avoids the issues which arise from maintaining a joint distribution over the data domain as the number of features increases, all while being able to generate privacy-preserving datasets. GEM+: While GEM offers better scalability than the prior methods from the literature, it cannot scale to the real-world high dimensionality demands and also suffers from lower fidelity compared to the more recent methods, such as AIM [McKenna et al., 2022]. Conversely, AIM offers high fidelity but fails to scale to high dimensions due to the memory

constraints of graphical models. To tackle this issue, GEM+ [Maddock et al., 2025] proposes a novel model which integrates the adaptive selection strategy of AIM with GEM's scalable generative networks, while also bringing algorithmic improvements specific to high-dimensional data.

**Kernel-based NN Models.** DP-MERF: Part of the line of frameworks that develop differentially private algorithms for training deep generative models is the private mean embeddings with random features (DP-MERF) method proposed by Harder et al. [2021]. Specifically, DP-MERF uses random Fourier features to approximate the kernel, which in turn allows for releasing a differentially private mean embedding of the real data. Given this differentially private embedding, the objective of the model's generator does not have direct access to the data and can then be freely optimised. DP-NTK: Building on DP-MERF, Yang et al. [2024] proposes a model which utilises the features of an empirical neural tangent kernel rather than the random Fourier features used by DP-MERF. This modification enables the model to better capture complex, non-linear relationships in tabular data, improving the balance between privacy and utility over the original DP-MERF, without relying on any public tabular datasets.

**Models adapted from other domains to tabular data synthesis.** In (§3.1-3.5) and earlier in this section, we compared deep generative models originally proposed for tabular data synthesis. Here, we discuss seminal models first developed in other fields and later adapted for tabular data. WGAN: Originally designed for computer vision tasks, WGAN [Arjovsky et al., 2017] was introduced in a paper that analyses ways to measure the distance between the real and model distributions, arguing that weaker loss functions induce weaker model topologies, thus weak models for the real distribution. The authors compare the Wasserstein distance with traditional probability distance measures, including Kullback-Leibler (KL) divergence, Jensen-Shannon divergence, and total variation distance, showing that the Wasserstein distance has properties that would be better suited for learning distributions in low dimensional manifolds. They thus propose GAN models that use the Wasserstein distance as loss function. WGAN-GP: A popular extension of WGAN is WGAN-GP [Gulrajani et al., 2017], which added a new term in the loss that penalised the gradient norm of the discriminator w.r.t. the discriminator's input, providing an alternative to the weight clipping that was leading to the generator retaining the real data points and adding Gaussian noise to fill the gaps. VEEGAN: VEEGAN [Srivastava et al., 2017] jointly trains a generator and a novel reconstructor network in order to avoid mode collapse. The reconstructor learns to map the real data distribution to Gaussian random noise, approximating the inverse of the generator and encouraging the generator to map the noise distribution back to the real data distribution.

### 4.7 Non-deep learning alternatives

While deep learning approaches for synthesising tabular data have gained significant traction, it is important to note that they are not always better than the statistical methods. Indeed, in certain scenarios, such methods can deliver competitive utility performance with lower computational costs and easier hyper-parameter tuning. Further, for use cases requiring strong privacy guarantees, non-deep learning models are often better at balancing the utility-privacy trade-off than their deep learning counterparts.

One example of a classical statistical model is Synthpop [Nowok et al., 2016], which is able to preserve multivariate relationships by modelling each tabular data feature conditional on the previously synthesised feature values. Generating synthetic data sequentially in this way avoids the training instability often seen with GAN-based models. More recently, Watson et al. [2023] introduced Adversarial Random Forests (ARF), which are tree-based generators able to combine the adversarial functionality of GANs with the robustness of tree ensembles to ultimately produce high-utility data. Yet another application domain where non-deep learning models excel is the privacy-preserving synthesis. For instance, rather than trying to approximate the full joint distribution of the real data, the Adaptive and Iterative Mechanism (AIM) [McKenna et al., 2022] directly optimises for pre-defined statistical queries and adds noise to a set of marginals of the data in order to satisfy differential privacy, and finally generates samples that best explain the marginals.

Finally, there are also works that bridge generative modelling and tree-based methods. For instance, Jolicoeur-Martineau et al. [2024] show that methods traditionally relying on deep learning, such as score-based diffusion and flow-matching, can be implemented using XGBoost instead of neural networks. This approach shows improved statistical fidelity and diversity compared to deep learning baselines such as Tab-

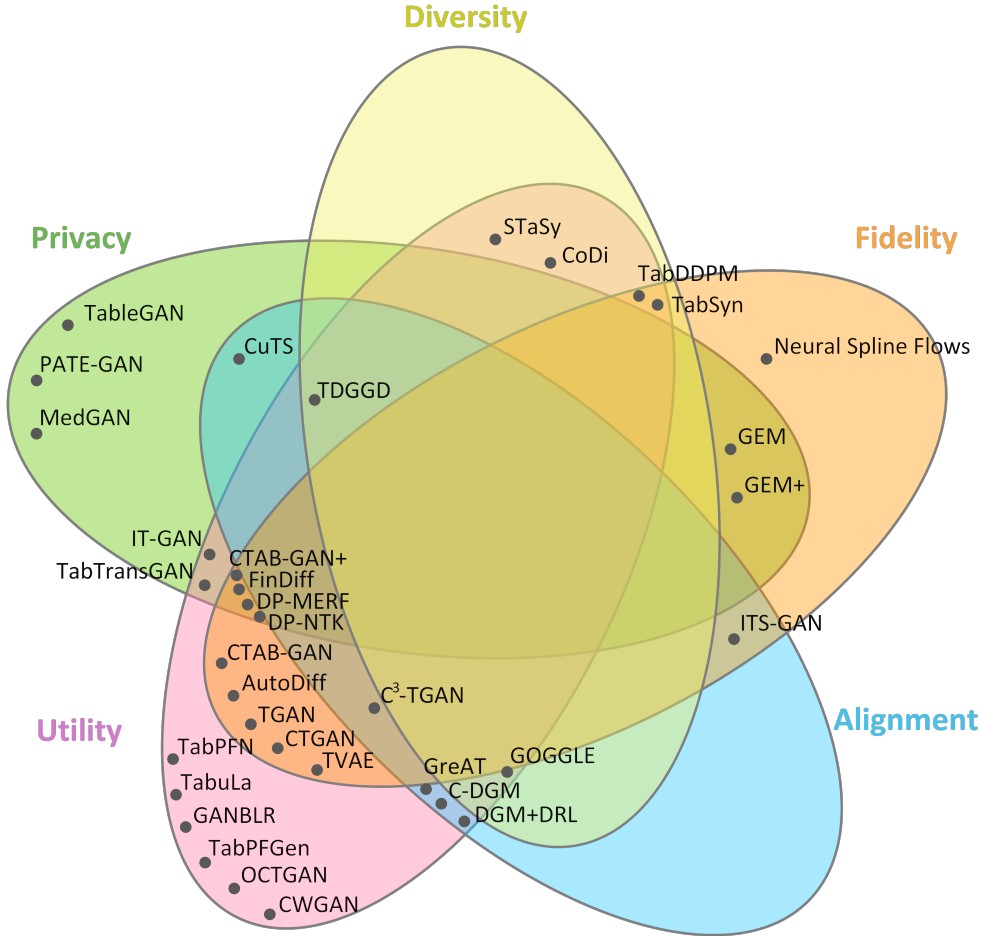

Figure 3: Visualisation of the surveyed models based on the requirements they address.

DDPM Kotelnikov et al. [2023] particularly on small datasets. Building on this approach, Cresswell & Kim [2024] address the scalability limitations of these tree-based models, proposing efficient implementations that enable scaling to larger datasets with a larger number of features and samples.

## 5    Discussion and Future Directions

In this survey, we identified five key requirements for effective synthetic data deployment, alongside relevant evaluation procedures and deep generative models, while categorising the latter by model architecture.

**Related work.** Previous surveys generally compare methods by the architecture-specific properties of the models. For example, both Borisov et al. [2024] and Davila R. et al. [2025] group methods mainly by their architectures, while also describing the necessary data transformation and regularisation techniques. [Lautrup et al., 2024] provide an overview of the models proposed in the field and the metrics used to evaluate them, while not taking this requirement-centric perspective which helps both newcomers and seasoned practitioners and researchers in the field to find the perfect model and evaluation protocol for their needs. Further, they exclude diffusion-based and LLM-based architectures. On the other hand, Wang et al. [2025] narrow the scope of their survey and focus specifically on methods for social and government sector applications, where balancing utility and privacy is the key concern.

Another related work is the survey by Jordon et al. [2022], which provides a valuable, high-level view of synthetic data across various modalities, including audio, image, video, time-series, and tabular data,

without focusing specifically on the latter. However, with a primary emphasis on privacy, their survey treats requirements such as utility and fidelity as secondary goals. Further, rather than reviewing in detail methods for generating the data, the authors offer a high-level description of what synthetic data is and why it is useful to society, arguing, alongside Savage [2023], that a general public understanding of these methods is crucial given their potential to contribute to data democratisation. While Jordon et al. [2022] advocate for systematic frameworks, they categorise methods primarily by application (e.g., private data release, fairness). Further, regarding tabular data, their work relies on GAN-based approaches and does not cover recent advancements such as diffusion or transformer-based architectures. Finally, van Breugel et al. [2024] review methods for improving data fairness and reducing bias. However, their work focuses on biomedically-relevant data (including molecular, tabular, and imaging data) and surveys foundation models able to create such data via query-specific generation. Thus, rather than focusing on methods that generate full datasets intended to replace or augment real data (i.e., the focus of this survey), they review methods which generate single samples conditioned on specific inputs (e.g., an answer in the form of an image to a medical question).

In contrast to these works, our survey distinguishes itself by providing a comprehensive overview of the requirements specific to synthesising tabular data and their complex interdependencies. To this end, we offer a practical, high-level roadmap (Figure 1) to guide users in selecting appropriate models. We support this roadmap with granular decision-making tools, including: (i) a structured technical taxonomy offering architectural details (Section 4), (ii) method-related specifics such as mixed data type support, inference time, and scalability to high-dimensional data (as part of Table 1), and (iii) an overview on which requirements are addressed by each method (Figure 3). The latter is particularly valuable for refining model selection based on requirements that are too intertwined to be effectively captured in the initial high-level roadmap. Furthermore, we support this user-centric perspective by detailing evaluation protocols and metrics for each requirement (Section 3). Overall, this structure positions our survey as a resource for both newcomers and specialists to select and evaluate models tailored to their specific application scenarios.

**Future Directions.** Tabular data generation is a rapidly evolving field still in its early stages. As illustrated in Figure 3, many early approaches have prioritised individual objectives—typically utility or privacy—leading to highly specialised models that lack broad applicability. Note that the categorisation in Figure 3 should be interpreted as a reflection of each method's focus, rather than an exclusive capability. As discussed in Section 3.6, these requirements are deeply interconnected: they often overlap or trade-off against one another. For example, the methods associated solely with the privacy requirement are positioned as such because their core contribution is a privacy-preserving mechanism. However, this does not implicitly suggest they lack utility (indeed, a synthesiser with very low utility would be useless in any downstream task), but rather that their innovations are driven by primarily addressing the privacy requirement. Similarly, the methods associated to the fidelity requirement focus on approximating the real data distribution, which often sets the foundation for achieving a high utility.

This fragmentation highlights a key challenge: the need for models that can balance multiple, often competing, requirements. Notably, alignment has historically received limited attention but is now emerging as a crucial factor for generating realistic data. With the increasing deployment of generative models in real-world applications and the growing prominence of neurosymbolic AI and safe AI [d'Avila Garcez et al., 2019; Giunchiglia et al., 2022], we anticipate that alignment will soon become a fundamental requirement in the field. Emerging approaches are already tackling this challenge. For instance, training-based methods like Stoian & Giunchiglia [2025] introduce new layers to guarantee constraint satisfaction and improve utility, while alternative directions focus on developing methods that do not require retraining, such as the work in Scassola et al. [2025b], which proposes a zero-shot framework utilising soft constraints to approximate the conditional distribution. Moreover, the Figure reveals a striking gap: no existing model successfully integrates all the core requirements we consider. As the field matures, we expect new requirements will emerge, and the hybridisation of techniques will drive the development of more versatile models. Bridging these gaps will be essential for advancing the field and enabling broader, more reliable applications of tabular data generation, potentially expanding to complex multi-table relational data settings as recently explored by Scassola et al. [2025a].

## Broader Impact Statement

This survey captures current trends in deep generative modelling for tabular data synthesis, offers a new perspective on what makes synthetic data practically useful, and outlines key steps toward achieving such a goal. We emphasise that, while existing methods address different requirements (e.g., alignment, utility, privacy, and fidelity), further progress toward developing approaches that simultaneously meet multiple requirements will increase their impact in real-world applications. In particular, we highlight the importance of incorporating domain knowledge to enhance the quality of synthetic data and encourage its integration in future deep generative modelling approaches for synthesising tabular data. It is important, however, to ensure that such background knowledge does not embed user-specific biases or characteristics, as this could raise privacy concerns. Provided this condition is met, injecting domain knowledge into generative models does not pose any direct negative impact.

## Acknowledgments

Mihaela Cătălina Stoian was supported by the EPSRC under the grant EP/T517811/1.

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

## A Appendix

In this Appendix, we provide the formulae for each of the metrics presented in Table 2. We denote by $N$ and $D$ the sampled population size (i.e., the total number of rows in a dataset) and the number of features (i.e., the total number of columns in a dataset), respectively. Moreover, $\mathbb{I}(\cdot)$ is the indicator function, which returns 1 if the condition inside the parentheses is true, and 0 otherwise, We also denote the *true positives*, *true negatives*, *false positives*, and *false negatives* by $TP$, $TN$, $FP$, and $FN$, respectively. The sets $\mathcal{R}$ and $\mathcal{S}$ represent the real and synthetic datasets, respectively.

**Utility**

$$\text{Accuracy} = \frac{TP + TN}{TP + TN + FP + FN} \tag{1}$$

$$\text{F1-score} = \frac{2 \cdot \text{Precision} \cdot \text{Recall}}{\text{Precision} + \text{Recall}}, \tag{2}$$

where $\text{Precision} = \frac{TP}{TP+FP}$ and $\text{Recall} = \frac{TP}{TP+FN}$.

$$\text{Root mean square error (RMSE)} = \sqrt{\frac{1}{N} \sum_{i=1}^{N} (y_i - \hat{y}_i)^2}, \tag{3}$$

where $y_i$ and $\hat{y}_i$ denote the actual (i.e., ground-truth) target value and the predicted target value for the $i$-th sample, respectively.

$$\text{Explained variance} = 1 - \frac{\text{Var}(\mathbf{y} - \hat{\mathbf{y}})}{\text{Var}(\mathbf{y})}, \tag{4}$$

where $\mathbf{y}$ and $\hat{\mathbf{y}}$ represent arrays of actual and predicted values, respectively, and $\text{Var}(\cdot)$ represents the variance of a set of values.

$$\text{Epistemic parity} = \frac{1}{|\mathcal{U}|} \sum_{u \in \mathcal{U}} \mathbb{I}[f(\mathcal{S}_u)) = f(D)], \tag{5}$$

where $\mathbb{D}$ is the universe of datasets, $f$ is a finding function $\mathbb{D} \to \{0, 1\}$, and $\mathcal{S}_u$ represents the synthetic dataset obtained by training a generator with seed $u$ drawn from the set of random seeds $\mathcal{U}$.

**Alignment**

$$\text{Constraint violation rate (CVR)} = \frac{1}{N} \sum_{i=1}^{N} \mathbb{I}(\text{sample } i \text{ violates any constraint}) \tag{6}$$

$$\text{Constraint violation coverage (CVC)} = \frac{\#\text{constraints violated by any of the N samples}}{\#\text{constraints}} \tag{7}$$

$$\text{Sample-wise constraint violation coverage (sCVC)} = \frac{1}{N} \sum_{i=1}^{N} \frac{\#\text{constraints violated by sample } i}{\#\text{constraints}} \tag{8}$$

**Fidelity**  In what follows, (i) $F$ represents the cumulative distribution function of a given feature in a dataset $\mathcal{D}$, (ii) $D_{KL}(\cdot||\cdot)$ denotes the Kullback-Leibler (KL) Divergence, measuring how one probability distribution diverges from a second, while $P$ and $Q$ denote the probability distributions of the real and synthetic data, respectively, (iii) $M$ is the mixture (average) distribution between $P$ and $Q$, and (iv) $\theta$ represents the parameters of the model trained to synthesise data.

Further, $\sup_x$ is the supremum (i.e., the least upper bound of a set of values over all possible $x$), and $I_\mathcal{S}(i,j)$ represents the mutual information between features $i$ and $j$ in dataset $\mathcal{D}$. Finally, $\rho_{\mathcal{S},ij}$ is the correlation coefficient (e.g., Pearson) between features i and j in dataset $\mathcal{D}$.

1. Feature-wise evaluation metrics:

$$\text{Wasserstein distance} = \int_{-\infty}^{\infty} |F_R(x) - F_S(x)| dx \tag{9}$$

$$\text{Kolmogorov-Smirnov test statistic} = \sup_x |F_R(x) - F_S(x)| \tag{10}$$

$$\text{Jensen-Shannon divergence} = \frac{D_{KL}(P||M) + D_{KL}(Q||M)}{2} \tag{11}$$

$$\text{Total variation distance} = \sum_x \frac{|P(x) - Q(x)|}{2} \tag{12}$$

2. Pair-wise evaluation metrics:

$$\text{RMSE differences in mutual information} = \sqrt{\frac{1}{D^2} \sum_{i,j} (I_\mathcal{R}(i,j) - I_\mathcal{S}(i,j))^2} \tag{13}$$

$$\text{MAE differences in mutual information} = \frac{1}{D^2} \sum_{i,j} |I_\mathcal{R}(i,j) - I_\mathcal{S}(i,j)| \tag{14}$$

$$\text{Correlation difference} = \frac{2}{D(D-1)} \sum_{i<j} |\rho_{\mathcal{R},ij} - \rho_{\mathcal{S},ij}| \tag{15}$$

3. Joint evaluation metrics:

$$\text{Likelihood fitness score} = \frac{1}{|\mathcal{R}|} \sum_{\mathbf{x} \in \mathcal{R}} \log p_\theta(\mathbf{x}) \tag{16}$$

$$\alpha\text{-Precision} = \frac{1}{|\mathcal{S}|} \sum_{\mathbf{x}_s \in \mathcal{S}} \mathbb{I}\left(\mathbf{x}_s \in \text{supp}_\alpha(\mathcal{R})\right), \tag{17}$$

where $\text{supp}_\alpha(\mathcal{R})$ is the minimum volume subset of $\text{supp}(\mathcal{R})$ that supports a probability mass of $\alpha$.

$$\text{Density} = \frac{1}{|\mathcal{S}|} \sum_{\mathbf{x}_s \in \mathcal{S}} \mathbb{I}\left(\exists \mathbf{x}_r \in \mathcal{R} \text{ s.t. } \|\mathbf{x}_s - \mathbf{x}_r\| \leq \text{NND}_k(\mathbf{x}_r)\right), \tag{18}$$

where $\text{NND}_k(\mathbf{x}_r)$ denotes the distance from $\mathbf{x}_r$ to the $k$-th nearest neighbour of $\mathbf{x}_r$.

$$\text{Clipped density} = \min\left(\frac{\text{Clipped density}_{\text{unnorm}}}{\text{Clipped density}_{\text{real}}}, 1\right), \tag{19}$$

where:

$$\text{Clipped density}_{\text{unnorm}} = \frac{1}{|\mathcal{S}|} \sum_{\mathbf{x}_s \in \mathcal{S}} \min\left(\frac{1}{k} \sum_{\mathbf{x}_r \in \mathcal{R}} \mathbb{I}(\|\mathbf{x}_s - \mathbf{x}_r\| \leq R_k(\mathbf{x}_r)), 1\right), \quad (20)$$

$$\text{Clipped density}_{\text{real}} = \frac{1}{|\mathcal{R}|} \sum_{\mathbf{x} \in \mathcal{R}} \min\left(\frac{1}{k} \sum_{\mathbf{x}_r \in \mathcal{R}\backslash\{\mathbf{x}\}} \mathbb{I}(\|\mathbf{x} - \mathbf{x}_r\| \leq R_k(\mathbf{x}_r)), 1\right), \quad (21)$$

and the clipped radius $R_k(\mathbf{x}_r)$ is defined as the minimum of $\text{NND}_k(\mathbf{x}_r)$, which is the distance from $\mathbf{x}_r$ to its $k$-th nearest neighbour, and the median of such distances across the dataset:

$$R_k(\mathbf{x}_r) = \min\left(\text{NND}_k(\mathbf{x}_r), \text{median}(\{\text{NND}_k(\mathbf{x}_r') : \mathbf{x}_r' \in \mathcal{R}\})\right) \quad (22)$$

**Privacy**

$$\text{AUCROC (for membership attacks)} = \mathbb{P}(\text{score(member)} > \text{score(non-member)}) \quad (23)$$

$$\text{Precision (for attribute disclosure attacks)} = \frac{TP}{TP + FP} \quad (24)$$

$$\text{Recall/Sensitivity (for attribute disclosure attacks)} = \frac{TP}{TP + FN} \quad (25)$$

$$\text{Distance to the closest record (DCR)} = \min_{\mathbf{x}_r \in \mathcal{R}} \text{dist}(\mathbf{x}_s, \mathbf{x}_r), \text{ for } \mathbf{x}_s \in \mathcal{S} \quad (26)$$

$$\text{Nearest neighbour distance ratio (NNDR)} = \frac{\min_{\mathbf{x}_r \in \mathcal{R}} \text{dist}(\mathbf{x}_s, \mathbf{x}_r)}{\min_{\mathbf{x}_r' \in \mathcal{R}\backslash\{\mathbf{x}_r\}} \text{dist}(\mathbf{x}_r, \mathbf{x}_r')}, \text{ for } \mathbf{x}_s \in \mathcal{S} \quad (27)$$

$$\text{k-anonymity} = \frac{1}{|\mathcal{R}|} \sum_{\mathbf{x}_r \in \mathcal{R}} \mathbb{I}(|\{\mathbf{x}' : v_{QI}^{\mathbf{x}'} = v_{QI}^{\mathbf{x}_r}\}| \geq k), \quad (28)$$

where $v_{QI}^{\mathbf{x}}$ represents the values of the quasi-identifier attributes (i.e., attributes that can be combined to uniquely identify an individual) for a given sample $\mathbf{x}$.

**Diversity**

$$\beta\text{-Recall} = \frac{1}{|\mathcal{R}|} \sum_{\mathbf{x}_r \in \mathcal{R}} \mathbb{I}(\mathbf{x}_r \in \text{supp}_\beta(\mathcal{S})), \quad (29)$$

where $\text{supp}(\mathcal{S})$ is the support of $\mathcal{S}$ and $\text{supp}_\beta(\mathcal{S})$ is the minimum volume subset of $\text{supp}(\mathcal{S})$ that supports a probability mass of $\beta$.

$$\text{Coverage} = \frac{1}{|\mathcal{R}|} \sum_{\mathbf{x}_r \in \mathcal{R}} \mathbb{I}(\exists \mathbf{x}_s \in \mathcal{S} \text{ s.t. } \|\mathbf{x}_r - \mathbf{x}_s\| < \text{NND}_k(\mathbf{x}_r)), \quad (30)$$

where $\text{NND}_k(\mathbf{x}_r)$ denotes the distance from $\mathbf{x}_r$ to the $k$-th nearest neighbour of $\mathbf{x}_r$.

$$\text{Clipped Coverage} = g\left(\frac{1}{|\mathcal{R}|} \sum_{\mathbf{x}_r \in \mathcal{R}} \min\left(\frac{1}{k} \sum_{\mathbf{x}_s \in \mathcal{S}} \mathbb{I}(\|\mathbf{x}_r - \mathbf{x}_s\| \leq \text{NND}_k(\mathbf{x}_r)), 1\right)\right), \quad (31)$$

where $g(\cdot)$ is a calibration function such that the metric scales linearly with the proportion of valid samples (see [Salvy et al., 2025]).

