# OpenReview forum: "A Survey on Deep Learning Approaches for Tabular Data Generation: Utility, Alignment, Fidelity, Privacy, Diversity, and Beyond"
_TMLR — Accepted by TMLR_

### Review · Reviewer_B4TN · 2025-10-31

**Summary Of Contributions:**

The paper is a survey of deep learning methods for synthetic tabular data generation, and metrics to evaluate these methods. Both metrics and methods are grouped by the goal they aim for. The paper considers four goals: utility, alignment, fidelity and privacy. Utility is measured by the performance of downstream predictive models. Alignment refers to domain-specific constraints the synthetic data should follow. Fidelity is measured by the statistical similarity of the synthetic and real data, or similarities of their marginals. Privacy is the vulnerability of the synthetic data to various attacks such as membership inference or attribute disclosure that undermine the privacy of the data subjects.

**Additional Comments:**

References:
- Carlini et al. (2022) "Membership Inference Attacks From First Principles" IEEE Symposium on Security and Privacy
- Harder et al. (2021) "DP-MERF: Differentially Private Mean Embeddings with Random Features for Practical Privacy-preserving Data Generation" AISTATS
- N. Hollman et al. (2025) "Accurate predictions on small data with a tabular foundation model" Nature
- T. Liu et al. (2021) "Iterative Methods for Private Synthetic Data: Unifying Framework and New Methods" NeurIPS
- McKenna et al. (2022) "AIM: an Adaptive and Iterative Mechanism for Differentially Private Synthetic Data" Proceedings of the VLDB Endowment
- M. F. Naeem, S. J. Oh, Y. Uh, Y. Choi, J. Yoo (2020) "Reliable Fidelity and Diversity Metrics for Generative Models" ICML
- Nowok et al. (2016) "synthpop: Bespoke Creation of Synthetic Data in R" Journal of Statistical Software
- I. Perez et al. (2024) "Does Differentially Private Synthetic Data Lead to Synthetic Discoveries?" Methods of Information in Medicine
- L. Rosenblatt et al. (2023) "Epistemic Parity: Reproducibility as an Evaluation Metric for Differential Privacy" Proceedings of the VLDB Endowment
- Salvy et al. (2025) "Enhanced Generative Model Evaluation with Clipped Density and Coverage" arXiv:2507.01761
- Yang et al. (2024) "Differentially Private Neural Tangent Kernels (DP-NTK) for Privacy-Preserving Data Generation" Journal of Artificial Intelligence Research
- Watson et al. (2023) "Adversarial Random Forests for Density Estimation and Generative Modeling" AISTATS

**Audience:**

Yes

**Audience Explanation:**

The paper is a decent survey of deep synthetic data generation methods for tabular data. Many surveys on synthetic data generation exist, but I'm not aware of one that looks at tabular data in general, including both generation and evaluation methods. The paper's categorisation based on the goal of of the generator or metric is also both novel and useful.

**Broader Impact Concerns:**

No ethical concerns.

**Claims And Evidence:**

No

**Claims Explanation:**

The paper is missing discussion on some important works. The scope defined in the paper is narrow in some aspects, and excludes discussion that would be beneficial for readers, especially those not familiar with tabular synthetic data generation. See Requested Changes for details.

**Requested Changes:**

**Important Changes**

The definition of utility in the paper (predictive performance) is narrow. Synthetic data could be used for many other purposes than machine learning. For example, Rosenblatt et al. (2023) and Perez et al. (2024) evaluate whether statistical inferences can be reproduced from synthetic data. These should be mentioned.

The focus on deep models is limiting. It is understandable to limit the scope, but there are several very good non-deep tabular data generators, especially with differential privacy, that are hidden by this focus. Examples include ARF (Watson et al. 2023), Synthpop (Nowok et al. 2016) and AIM (McKenna et al. 2022). While these do not need to be discussed in the same detail as the deep learning models, it would be beneficial to tell readers that deep learning models are not always the best tabular data generators, and give some pointers to alternatives.

In-scope methods or evaluation metrics that should be included:
- TPR at low FPR (Carlini et al. 2022) should be mentioned as a MIA evaluation metric.
- There is a line of work on synthetic data fidelity and diversity metrics that is not discussed. See Salvy et al. (2025) and the metrics they compare with. The original papers focus on image data, but these metrics can be used with tabular data, and have been used by for example Kotelnikov et al. (2023).
- TabPFNv2 (Hollman et al. 2025) can generate synthetic tabular data.
- GEM (Liu et al. 2021), DP-MERF (Harder et al. 2021) and derivatives of DP-MERF (Yang et al. 2024 and their references) are differentially private deep generative models.

**Minor changes**
- Not clear what "Generation Time" column in Table 2 is based on, and precisely what "slow", "fast" etc. mean.
- The mathematical description of a tabular dataset is unusual and hard to understand. A more understandable description would be defining datapoints as tuples, and a dataset as a list (or vector) of tuples.
- The definition $X \in \cup\_{j=1}^K \mathbb{D}^j$ in the beginning of page 2 does not make sense with the surrounding definitions. A datapoint is a tuple where each element is from one of the $\mathbb{D}^j$ sets, not a single element from the union.
- The term "fidelity" has an established but different meaning in synthetic data evaluation literature (for example Naeem et al. 2020).
- On the alignment definition: it is possible for real data to contain erroneous values. In the given example, the minimum and maximum haemoglobin levels could swapped due to human error, so the statement that real data do not contain such values is not always correct.
- Membership attacks are usually called membership inference attacks.
- Avoid rotated tables if at all possible. They are annoying to read.
- Table 2 should be closer to the introduction where it is referenced.
- Rows in Table 1 are not horizontally aligned, for example compare the second and third columns of the fourth row.

---

> ### Author Response · Authors · 2025-12-02
>
> We sincerely thank the reviewer for their positive assessment of our work, particularly for recognising the "novel and useful" categorisation. We also appreciate the constructive feedback regarding the scope of our utility evaluation protocol and the inclusion of a discussion on non-deep learning methods. We have revised the manuscript to incorporate these suggestions  (using colors to highlight specific changes).
>
> ### 1. Utility
>
> > **Feedback 1.A. (Regarding utility):**  *Synthetic data could be used for many other purposes than machine learning. For example, Rosenblatt et al. (2023) and Perez et al. (2024) evaluate whether statistical inferences can be reproduced from synthetic data. These should be mentioned.*
>
> **Response 1.A.:**
> We have broadened the utility evaluation protocol in **Section 3.1** (in light blue text) and we have explicitly incorporated the suggested references.
> Specifically, we emphasised that synthetic data must support valid inference (to ensure that statistical conclusions derived from real data are preserved), and not just predictive accuracy for machine learning models.
> To this end, we described the epistemic parity metric (proposed by Rosenblatt et al. (2023)) and  noted the importance of  evaluating
> the validity and power of statistical tests, especially  in the context
> of synthesising data intended for scientific discovery.
>
>
> ### 2. Missing Methods
>
> > **Feedback 2.A. (Regarding non-deep learning models):**
> *The focus on deep models is limiting. It is understandable to limit the scope, but there are several very good non-deep tabular data generators, especially with differential privacy, that are hidden by this focus. Examples include ARF (Watson et al. 2023), Synthpop (Nowok et al. 2016) and AIM (McKenna et al. 2022). While these do not need to be discussed in the same detail as the deep learning models, it would be beneficial to tell readers that deep learning models are
> not always the best tabular data generators, and give some pointers to alternatives.*
>
> **Response 2.A.:**
> We have added **Section 4.7 (Non-deep learning alternatives)** (in light blue text) to address this, where we included a brief  discussion on the suggested methods (Synthpop, ARF, and AIM).
> In particular, we highlighted that for scenarios requiring rigorous privacy guarantees (AIM), these methods often offer competitive performance to deep generative models, especially for low-dimensionality data.
>
> > **Feedback 2.B. (Regarding additional deep learning models):**
> *In-scope methods or evaluation metrics that should be included: TabPFNv2 (Hollman et al. 2025), GEM (Liu et al. 2021), DP-MERF (Harder et al. 2021) and derivatives of DP-MERF (Yang et al. 2024).*
>
> **Response 2.B.:**
> We thank the reviewer for suggesting to include these methods.
> We have significantly expanded the survey to include all the requested models in **Section 4** (in light blue text):
> * **GEM** is reviewed under **Iterative Privacy-Preserving NN Models** (Section 4.6).
> * **DP-MERF** and **DP-NTK** (Yang et al., 2024) are reviewed under **Kernel-based NN Models** (Section 4.6).
> * **TabPFN** is reviewed under **Transformer-based Methods** (Section 4.3), highlighting its capabilities as a foundation model.
> * Additionally, we reviewed the recently proposed **GEM+**[1] model, under **Iterative Privacy-Preserving NN Models** (Section 4.6).
>
> This change is also reflected in Figures 1 and 3, as well as Table 1 (all of which now include these models).
>
> **References**
>
> [1] Samuel Maddock, Shripad Gade, Graham Cormode, and Will Bullock. GEM+: scalable state-of-the-art private synthetic data with generator networks. CoRR, abs/2511.09672, 2025.
>
>
>
> > **Feedback 2.C. (Regarding additional metrics):**
> *TPR at low FPR (Carlini et al. 2022) should be mentioned as a MIA evaluation metric. There is a line of work on synthetic data fidelity and diversity metrics that is not discussed. See Salvy et al. (2025) and the metrics they compare with. The original papers focus on image data, but these metrics can be used with tabular data, and have been used by for example Kotelnikov et al. (2023).*
>
>
> **Response 2.C.:**
> We have updated **Section 3.4 (Privacy)** (in light blue text) to discuss the limitations of aggregate metrics like AUCROC. We now explicitly cite Carlini et al. (2022) and recommend **TPR at low FPR** for detecting high-confidence privacy leakage.
>
> Additionally, we added a fifth requirement, diversity, in Section 3.5 (in blue text), and updated Table 2 accordingly.
> In this section, we also discussed popular metrics to evaluate diversity, such as $\beta$-recall, authenticity, and coverage.
> We also included (in light blue text) clipped coverage (recently proposed by Salvy et al. (2025), ensuring a more robust evaluation of diversity compared to the standard metrics.

---

> > ### Author Response · Authors · 2025-12-02
> >
> > ### 3. Other changes
> >
> > > **Feedback 3.A.:**  *Not clear what "Generation Time" column in Table 2 is based on.*
> >
> > **Response 3.A.:**
> > We added a clear definition for the "Generation Time" thresholds (fast/medium/slow) in the caption of **Table 1**, as well as a footnote (on page 2) to further clarify how we collected the  generation time values, which we ultimately used to decide the thresholds.
> >
> >
> > > **Feedback 3.B.:**  *Avoid rotated tables if at all possible. They are annoying to read.*
> >
> > **Response 3.B.:**
> > We changed the formatting of **Table 1** (by reducing column separation) to fit it horizontally, and removed the rotation.
> > However, we kept the rotation on Table 2 (the metrics table) due to lack of space.
> >
> >
> >
> > > **Feedback 3.C.:**  *Membership attacks are usually called membership inference attacks.*
> >
> > **Response 3.C.:**
> > We revised the manuscript with this change.
> >
> >
> >
> > > **Feedback 3.D.:**  *Table 2 should be closer to the introduction where it is referenced.*
> >
> > **Response 3.D.:**
> > We revised the manuscript with this change (please note that Table 2 now corresponds to Table 1, page 4).
> >
> >
> > > **Feedback 3.E.:**  * Rows in Table 1 are not horizontally aligned, for example compare the second and third columns of the fourth row.*
> >
> > **Response 3.E.:**
> > We revised the manuscript with this change (please note that Table 1 now corresponds to Table 2, page 6).
> >
> > > **Feedback 3.F.:**  *The term "fidelity" has an established but different meaning in synthetic data evaluation literature (for example Naeem et al. 2020).*
> >
> > **Response 3.F.:**
> > In the context of tabular data, we define fidelity broadly as the preservation of statistical properties. However, we note that in the generative model literature (e.g., \citep{naeem2020reliable}), fidelity is often strictly defined as precision.
> >
> >
> > > **Feedback 3.G.:**  *On the alignment definition: it is possible for real data to contain erroneous values. In the given example, the minimum and maximum haemoglobin levels could swapped due to human error, so the statement that real data do not contain such values is not always correct.*
> >
> > **Response 3.G.:**
> > We included a discussion to address this problem and how to mitigate it in Section 3.6, under "Alignment and Fidelity".

---

> > > ### Comment · Reviewer_B4TN · 2025-12-10
> > >
> > > Thank you for the response. You have addressed most of my concerns, and the updated paper is much better. I still have some fairly minor suggestions:
> > > - Authenticity does not measure diversity. Completely memorising the real data should give perfect diversity, but no authenticity.
> > > - I didn't find any changes regarding the confusing notation for tabular data (2nd and 3rd bullet points in "Minor Changes" in my original review).
> > > - Table 1 is somewhat difficult to read due to its compactness. Grouping the rows in some way and adding lines to separate the groups could help. Another possibility is adding a bit of vertical space between rows.
> > > - Table 1 caption should specify where the data for generation time comes from.
> > > - Statistical utility should be discussed already in Section 2.
> > > - Epistemic parity should be in Table 2.
> > > - It looks like Table 2 would fit in normal orientation if the "Formula" column was removed. That column is useful for some metrics, but many of them are too complex to understand based on just one formula. Would it be possible to add an appendix section with short descriptions of each metric to replace that column?
> > > - Figure 2, constraints-enforcing panel: "Layer" should be renamed to highlight that it is enforcing the constraints.
> > > - In the GAN section, it is not clear why some GANs have separate paragraphs and some do not.

---

> ### Author Response · Authors · 2025-12-22
>
> We thank the reviewer for their continued engagement and insightful comments throughout this process. The additional feedback provided has been invaluable, and we believe incorporating these suggestions has directly resulted in a stronger version of our survey.
> Below we address the remaining concerns.
>
> > **Feedback 1. (Authenticity and Diversity):** *Authenticity does not measure diversity. Completely memorising the real data should give perfect diversity, but no authenticity.*
>
> **Response 1.:**
> We agree with the reviewer that including authenticity as a metric for sampling diversity may cause confusion, despite our initial reasoning that synthetic data must be authentic to be considered truly diverse.
> To this end, we removed authenticity from the list of diversity metrics  in both Section 3.5 and Table 2.
>
>
> > **Feedback 2.a. (Tabular Data Notation):** *I didn't find any changes regarding the confusing notation for tabular data (2nd and 3rd bullet points in "Minor Changes" in my original review). The mathematical description of a tabular dataset is unusual and hard to understand. A more understandable description would be defining datapoints as tuples, and a dataset as a list (or vector) of tuples.*
>
> **Response 2.a.:**
> We apologise for overlooking the specific response to this previous point (and point 2.b. addressed below) in our last revision. We appreciate this was brought back to our attention.
> We have now revised the mathematical description of tabular datasets (on page 3, with the change marked in green) to improve readability. As suggested, we now define data points as tuples and the dataset as an indexed collection of these tuples, clarifying that the $i$-th sample constitutes the $i$-th row.
>
>
> > **Feedback 2.b.:** *The definition in the beginning of page 2 does not make sense with the surrounding definitions. A datapoint is a tuple where each element is from one of the sets, not a single element from the union.*
>
> **Response 2.b.:**
> The reviewer is right. We updated the definition (again on page 3, with the change marked in green) to clarify that the unknown joint distribution $p_\mathcal{X}$ is over the random variables $X_j$, where $X_j \in \mathbb{D}^j$, such that a sample drawn from $p_\mathcal{X}$ is a tuple where the $j$-th element is from $\mathbb{D}^j$.
>
>
> > **Feedback 3. (Table 1 Readability):** *Table 1 is somewhat difficult to read due to its compactness. Grouping the rows in some way and adding lines to separate the groups could help. Another possibility is adding a bit of vertical space between rows.*
>
> **Response 3.:**
> In the revised manuscript, we increased the spacing between rows in Table 1 to improve readability. Further, we added a light grey background to alternating rows.
>
>
>
>
> > **Feedback 4. (Table 1 Caption):** *Table 1 caption should specify where the data for generation time comes from.*
>
> **Response 4.:**
> To make the information regarding generation times easier to locate, we moved the explanation from the footnote on page 2 to the caption of Table 1. The caption now explicitly specifies the source of the data generation times.
>
>
> > **Feedback 5. (Statistical Utility):** *Statistical utility should be discussed already in Section 2.*
>
> **Response 5.:**
> We updated Section 2 to include a brief statement (in green) regarding statistical utility in the context of differentially private settings.
> We provide a more detailed discussion of this in Section 3.1.

---

> > ### Author Response · Authors · 2025-12-22
> >
> > > **Feedback 6. (Epistemic Parity):** *Epistemic parity should be in Table 2.*
> >
> > **Response 6.:**
> > We have now updated Table 2 to include epistemic parity as a metric for measuring utility.
> > We also added the following metrics for measuring fidelity: $\alpha$-precision, density, and clipped density.
> >
> > > **Feedback 7. (Table 2 Formulae):** *It looks like Table 2 would fit in normal orientation if the "Formula" column was removed. That column is useful for some metrics, but many of them are too complex to understand based on just one formula. Would it be possible to add an appendix section with short descriptions of each metric to replace that column?*
> >
> > **Response 7.:**
> > We agree that the 'Formula' column made the table dense and difficult to read. Consequently, we have removed this column from Table 2 in the revised manuscript.
> > Further, as suggested, we moved the detailed formulae and terminology definitions to the newly added Appendix (highlighted in green).
> >
> >
> >
> > > **Feedback 8. (Figure 2 Terminology):** *Figure 2, constraints-enforcing panel: "Layer" should be renamed to highlight that it is enforcing the constraints.*
> >
> > **Response 8.:**
> > In Figure 2, within the "Constraints-enforcing model" panel, we renamed the "Layer" to "Constraints-enforcing Layer" to explicitly highlight its function.
> >
> > > **Feedback 9. (GAN Section Structure):** *In the GAN section, it is not clear why some GANs have separate paragraphs and some do not.*
> >
> > **Response 9.:**
> > We restructured the GAN section (i.e., Section 4.1 "Generative Adversarial Network-based Methods") into specific subsections to clearly distinguish between standard and hybrid architectures.
> > For instance, Section 4.1.1 ("GAN-based models") groups methods like CWGAN, TGAN, and CTGAN, which rely on standard GAN architectures.
> >
> > In contrast, Section 4.1.2 ("GAN and BN-based models") includes hybrid models like GANBLR and C$^3$-TGAN.
> > While these models operate within the adversarial generator-discriminator paradigm, they integrate Bayesian Networks (BNs) into their architecture.
> > Specifically, GANBLR constructs both its generator and discriminator using BNs, whereas C$^3$-TGAN uses BNs to model feature correlations and augment the generator's input.
> > This restructuring highlights that while they share the adversarial "skeleton", they fundamentally differ in their component design.
> >
> > Since similar distinctions exist within other paradigms discussed in Section 4 (e.g., diffusion-based, transformer-based architectures, etc.), we applied the same restructuring.
> > More precisely, we converted paragraphs into dedicated subsections for each of the subsequent Sections 4.2, 4.3, etc.
> > We believe that this change improves readability and visually clarifies the grouping of different synthesising models.

---

> > > ### Comment · Reviewer_B4TN · 2025-12-30
> > >
> > > Thank you for the new changes, the paper has been greatly improved from the first submission. I still have some minor comments on the new parts, but these are small enough that I'm already recommending acceptance.
> > >
> > > - End of page 4 and page 6: epistemic parity (or statistical utility in general) is not specific to differentially private synthetic data, even though Rosenblatt et al. focus on that. For example, early work on synthetic data (Raghunathan et al. 2003) considered statistical inference the main application of synthetic data.
> > > - Precision and Recall are not defined for Eq. (2).
> > > - After Eq. (4) "Var" is written in cursive.
> > > - Eq. (10): "test" -> "test statistic".
> > > - Eq. (18), (22): NND is not defined.
> > > - Eq. (24): "F" -> "FP".
> > >
> > > Reference:
> > > - Raghunathan, Trivellore E., Jerome P. Reiter, and Donald B. Rubin. "Multiple Imputation for Statistical Disclosure Limitation." Journal of Official Statistics (2003)

---

> ### Author Response · Authors · 2026-01-01
>
> We thank the reviewer for their positive appraisal and for acknowledging the improvements made during the discussion period. We genuinely appreciate the reviewer's insightful feedback, which we believe has directly led to a significantly stronger manuscript.
>
> We address the final concerns below.
>
>
>
>
> > **Feedback 1:**
> *End of page 4 and page 6: epistemic parity (or statistical utility in general) is not specific to differentially private synthetic data, even though Rosenblatt et al. focus on that. For example, early work on synthetic data (Raghunathan et al. 2003) considered statistical inference the main application of synthetic data.*
>
> **Response 1:**
> We have now updated the manuscript (beginning of page 5, and end of page 6) to mention   that statistical utility was the focus of foundational work (Raghunathan et al., 2003) on generating fully synthetic data using classical statistical methods, before describing the more recent works by Rosenblatt et al. (2023) and by Perez et al. (2024).
>
>
> > **Feedback 2:**
> *Eq. (18), (22): NND is not defined.*
>
> **Response 2:**
> We have updated the manuscript to include a definition of $\text{NND}_k(\mathbf{x})$ after Eq. (18).
> Further, we have also added this definition just before Eq. (22) to improve clarity.
>
>
> > **Feedback 3:**
> - *Precision and Recall are not defined for Eq. (2).*
> - *After Eq. (4) "Var" is written in cursive.*
> - *Eq. (10): "test" -> "test statistic".*
> - *Eq. (24): "F" -> "FP".*
>
> **Response 3:**
> We corrected these issues in the revised manuscript.

---

### Review · Reviewer_NnVd · 2025-11-11

**Summary Of Contributions:**

## Contributions

The paper reviews tabular synthetic data generation methods. They group these based on requirements "Utility", "Alignement", "Fidelity", and "Privacy", and later based on underlying modelling method.

## Strengths

1. I applaud the idea of a user-centric survey that could help practitioners generate and evaluate their models.
2. The method survey is quite extensive, and the in-text explanations of individual works generally good.
3. Although there are surveys out there, a new survey that keeps up-to-date with recent methods (e.g. in diffusion) is valuable.

## Weaknesses
### 1. Contribution and structure.

Authors write in the introduction:

> This survey takes a unique stance and analyses models and evaluations methods alike from the point of
view of the user, categorising them on the ground of the requirements they primarily address and measure:
utility, alignment, fidelity, and privacy

I do not agree that the current paper provides sufficient insight or recommendations to help users choose a model, generate data, or evaluate the data. For example, Figure 1 contains high-level ideas of models, but without additional explanations this figure is not comprehensible. Similarly, there are so many methods that a user could choose from---it would have been great if there was more structure in which they are presented (e.g. a decision tree for methods, or at least multiple axes on which different methods are compared such that the user can immediately see which methods would apply to their setting)

More importantly, I do not agree with the general structure of the paper. The Sections 3-6 (e.g. "Utility", "Alignment") suggest authors will focus on metrics/requirements, but then Section 3.2, 4.2, 5.2, discuss methods. It seems authors are of the opinion that generative models are designed for one metric (e.g. utility), but this is not true---methods hopefully optimize for multiple metrics (at least 1 and 3 at the same time, privacy is a separate objective and alignment can be targeted explicitly or implicitly). There are also trade-offs that make it unwise for users to focus on one requirement, ignoring another. This is also reflected in Table 2's indication that methods have a "Primary requirement". The authors do seem to acknowledge the overlap in Figure 2, but here too methods like "PATE-GAN" are not in the "fidelity/utility" circles---even though such methods are clearly designed to also be close to real data. Having separate Metrics and Methods sections would in my opinion be more appropriate (where methods can be split up according to Figure 2). This would also get rid of Section 7, which is a bit of a non-section as the methods could easily be applied to e.g. fidelity.


### 2. Metrics

Beyond the structure, the discussion of metrics is in general, insufficient. References are missing, in text as well as from Table 1. The "Alignment" requirement is strictly weaker than "Fidelity"---this is fine, but the relationship between the two should be highlighted. These relationships also consist of trade-offs, e.g. between "privacy" and "fidelity", which is not acknowledged.

There is also more work on joint-distribution fidelity, e.g. (Improved) Precision and Recall, Density and Coverage, or Alpha-Precision and Recall. These are more modern and well-cited, and I think they are valuable to capture high-dimensional data quality. At last, there is also a requirement that is not mentioned in the paper: diversity/coverage. This is important, e.g. for the mentioned example of generating "rare data".

Alaa, A. M., van Breugel, B., Saveliev, E., & van der Schaar, M. (2022). How Faithful Is Your Synthetic Data? Sample-Level Metrics for Evaluating and Auditing Generative Models. International Conference on Machine Learning (ICML), 290–306.

Gulrajani, I., Raffel, C., & Metz, L. (2019). Towards GAN Benchmarks Which Require Generalization. 7th International Conference on Learning Representations, ICLR 2019.

Kynkäänniemi, T., Karras, T., Laine, S., Lehtinen, J., & Aila, T. (2019). Improved Precision and Recall Metric for Assessing Generative Models. Advances in Neural Information Processing Systems, 32. https://github.com/kynkaat/improved-precision-and-recall-metric.

Naeem, M. F., Oh, S. J., Uh, Y., Choi, Y., & Yoo, J. (2020). Reliable Fidelity and Diversity Metrics for Generative Models. In H. D. III & A. Singh (Eds.), Proceedings of the 37th International Conference on Machine Learning (Vol. 119, pp. 7176–7185). PMLR. http://proceedings.mlr.press/v119/naeem20a.html

Sajjadi, M. S. M., Bachem, O., Lucic, M., Bousquet, O., & Gelly, S. (2018). Assessing Generative Models via Precision and Recall. In S. Bengio, H. Wallach, H. Larochelle, K. Grauman, N. Cesa-Bianchi, & R. Garnett (Eds.), Advances in Neural Information Processing Systems (Vol. 31).


### 3. Low-level structure

Besides the high-level structure, the article's structure can be improved more granularly. For example, Section 6.2.; starts with "AE and GAN-based models", then continues with "GAN-based models". Similar in Section 4.2, and 5.2 has similar issues (diffusion-based, followed by more diffusion paragraphs).

### 4. Contributions/related work

> [this survey] offers a new perspective on what makes synthetic data practically useful,

It would be helpful if the authors specify what exactly this new perspective is compared to existing surveys. They mention on p.12 some related work, but there are much more related surveys, e.g. (Jordon et al, 2022), (Savage et al, 2023) or (Van Breugel et al, 2024). I do see value in a new review, especially since the popularity of diffusion models (which these previous reviews hardly capture).

Jordon, J., Szpruch, L., Houssiau, F., Bottarelli, M., Cherubin, G., Maple, C., Cohen, S. N., & Weller, A. (2022). Synthetic Data—What, why and how? (No. arXiv: 2205.03257). https://doi.org/10.48550/arxiv.2205.03257

Savage, N. (2023). Synthetic data could be better than real data. Nature. https://doi.org/10.1038/D41586-023-01445-8

van Breugel, B., Liu, T., Oglic, D., & van der Schaar, M. (2024). Synthetic data in biomedicine via generative artificial intelligence. Nature Reviews Bioengineering, 2(12), 991-1004.


### 5. Writing

Writing could be improved throughout. Some examples:


>  [p.2] While above we have the goal for standard generative modelling,
we know that different use cases can lead to additional requirements, which lead to different goals for the
generative task.

Bit vague, please rephrase.


> [p.2] ..., i.e., stating which samples are admissible and which are not, the goal is to learn the parameters
θ of a generative model such that (i) the model distribution pθ approximates pX, and ...

Part "(i) the model distribution pθ approximates pX" is important for most requirements, so it can be removed here (it's already at the top of the page).


> [p.4] 3[.0] Utility

Utility is important because it's easy and targets the downstream task directly. This paragraph is quite unclear and I suggest focusing on the core---it's ease of use---instead of e.g. the tangent of rare data.

> [p.7] 5[.0] Fidelity

Again, these "Section.0" paragraphs can be shortened and more to the point as it's currently a bit unclear what the authors want to achieve with these paragraphs.

**Audience:**

No

**Audience Explanation:**

The paper has potential and could indeed be interesting to a large group of practitioners if the recommendations were specific (e.g. if their was a decision tree or very clear structure on which practitioners could base their generation and evaluation protocols). In the current form, however, it will be difficult for practitioners to understand what method to choose.

**Broader Impact Concerns:**

No concerns

**Claims And Evidence:**

No

**Claims Explanation:**

The grouping is the primary contribution, yet this grouping is not motivated, and in my opinion, incorrect (see Weakness 1).

**Requested Changes:**

See the weaknesses above. In particular, for me the following are vital to acceptance:
1) the trade-off between requirements
2) inclusion of diversity requirement
3) regrouping that acknowledges this trade-off. This probably means a general restructuring of the paper into metrics versus methods

Not essential but highly recommended
1) more literature on metrics
2) clearer structure for practitioners that can truly help them match their setting with the best method
3) cleaner paragraph structure and writing
4)

---

> ### Author Response · Authors · 2025-12-02
>
> We sincerely thank the reviewer for their detailed and constructive feedback, and for   recognising the value of a user-centric survey designed to help practitioners in generating and evaluating models. Your insights regarding the paper's structure, the categorisation of methods, and the missing "Diversity" requirement were particularly valuable. We have substantially revised the manuscript to address these points (using colors to highlight specific changes).
>
>
> ### 1. Contribution and structure
>
> > **Feedback 1.A. (Roadmap to help select a model):** *it would have been great if there was more structure in which they are presented (e.g. a decision tree for methods, or at least multiple axes on which different methods are compared such that the user can immediately see which methods would apply to their setting.*
>
> **Response 1.A.:**
> We created **Figure 1**, which is a practical **roadmap** designed to guide practitioners in selecting an appropriate synthesiser based on their specific needs (such as whether the data is sensitive, high-dimensional, requires fast inference, or needs domain constraints enforced). This roadmap directly references the surveyed models, and can be seen as a simplified visualisation of our survey.
>
> > **Feedback 1.B. (Restructuring and eliminating grouping by a primary requirement):**  *More importantly, I do not agree with the general structure of the paper. [..] Having separate
> Metrics and Methods sections would in my opinion be more appropriate (where methods can be split up according to Figure 2).*
>
> **Response 1.B.:**
> We thank the reviewer for their insightful comment, which allowed us to revise and improve the structure of our survey. To this end, we have restructured the survey into "Requirements and Evaluation Metrics" and "Methods" sections. This decoupling allows us to analyse how models address multiple requirements simultaneously, rather than categorising them by a single primary requirement.
> The paper now follows the suggested structure:
> * **Section 3 (Requirements and Evaluation Metrics):** This section details the five requirements (Utility, Alignment, Fidelity, Privacy, and Diversity) and the evaluation protocols and metrics for each of them.
> * **Section 4 (Methods):** This section surveys the methods grouped by their underlying model's architecture (GANs, Diffusion, Transformers, etc.).
>
> This change eliminates the issue of associating models to a single primary requirement. Instead, we now capture the overlap in **Table 1**, which uses a multi-column checkmark system to show that models address multiple requirements simultaneously (and which does not contain the "Primary Requirement" column anymore), as well as in *Figure 3* which visualises the surveyed models by a Venn diagram w.r.t. the requirements.
>
> > **Feedback 1.C. (Regarding Figure 3 interpretation and trade-offs between requirements):** *
> There are also trade-offs that make it unwise for users to focus on one requirement, ignoring another.
> The authors do seem to acknowledge the overlap in Figure 2, but here too
> methods like "PATE-GAN" are not in the "fidelity/utility" circles---even though such methods are clearly designed to also be close to real data.*
>
> **Response 1.C.:**
> In addition to eliminating the methods' grouping by a primary requirement, we have now added a clarification (in purple text) on how to interpret Figure 3 (please note that in the revision the Venn diagram does not correspond to Figure 2 anymore, but to Figure 3) in the context of the requirements, on page 17, under the "Future directions" paragraph.
>
> Further, we added a discussion on the relationships among the requirements (including trade-offs) in Section 3.6 (in blue text).

---

> ### Author Response · Authors · 2025-12-02
>
> (Response cont'd)
>
> ### 2. Metrics
>
> > **Feedback 2.A. (Diversity Requirement and Metrics):**
> *Beyond the structure, the discussion of metrics is in general, insufficient. [...] At last, there is also a requirement that is not mentioned in the paper: diversity/coverage. This is important, e.g. for the mentioned example of generating "rare data".*
>
> **Response 2.A.:**
> We have substantially expanded the discussion on metrics and added **Diversity** as a fifth core requirement in **Section 3.5** and further discussed the example of generating rare data in the context of diversity.
> * We also incorporated the suggested literature on evaluation metrics.
> * We added relevant metrics to **Table 2**, including **Coverage**, **$\beta$-Recall**, and **Authenticity**, explicitly discussing how they capture the variability of the data distribution.
>
>
> > **Feedback 2.B. (Relationships among the requirements):**  *The "Alignment" requirement
> is strictly weaker than "Fidelity"---this is fine, but the relationship between the two should be highlighted. These relationships also consist of trade-offs, e.g.
> between "privacy" and "fidelity", which is not acknowledged.*
>
> **Response 2.B.:**
> As mentioned above, we have added a dedicated subsection, **Section 3.6 (Relationships Among the Requirements)**, to explicitly discuss these interactions.
> * We analyse the hierarchy between  Fidelity and Alignment (noting  (i) that perfect fidelity implies alignment, but not vice-versa, (ii) that alignment is crucial when the real data has errors, and (iii) that methods guaranteeing alignment can be used to check the real data before it is used to train a synthesiser).
> * We discuss the inherent tension (trade-off) between **Privacy and Utility/Fidelity**, as well as the complementary relationship between **Fidelity** and **Diversity**.
>
>
> ### 3. Low-level structure and writing
>
> > **Feedback 3.A. (Regarding low-level structure):**  *Besides the high-level structure, the article's structure can be improved more granularly. For example, Section 6.2.; starts with "AE and GAN-based models", then continues with "GAN-based models". Similar in Section 4.2, and 5.2 has similar issues (diffusion-based, followed by more diffusion paragraphs).*
>
> **Response 3.A.:**
> The new structure in **Section 4 (Methods)** resolves these inconsistencies. We now group methods strictly by their architectural family (e.g., **4.1 GAN-based**, **4.2 Diffusion-based**, **4.3 Transformer and Large Language Model-based**, **4.6 Other Deep Learning Methods**). Within these sections, we clearly highlight hybrid approaches (e.g., "GAN and VAE-based") to avoid the confusing flow noted in the previous draft.
>
>
>
> > **Feedback 3.B. (Regarding writing):**
>
> **Response 3.B.:**
> We have revised the text throughout to improve clarity. Further, we have revised the introduction to clearly outline the importance of tabular data synthesis and its challenges (in blue text).

---

> > ### Comment · Reviewer_NnVd · 2025-12-11
> >
> > The authors have done a great job at improving the paper, e.g. I applaud the new structure, the new Figure 1, Section 3.6, and many small changes others and I have asked for.
> >
> > At the same time, I think the paper needs further improvements:
> > * Its main claim is that it offers a unique perspective, which it still does not. For example, Jordon et al. and others have similarly structured reviews of synthetic data by "use to practitioners" (my previous point 4, which has not been responded too). Though something like Figure 1 really goes in the direction of added value for practitioners, it could be significantly improved—e.g. more decision functions (e.g. based on a priori knowledge, number of samples, mixed type or not), more granular nodes (there are sometimes still 10 methods to choose from), and "fast inference" and "high-dimensional" is subjective (e.g. please quantify these in the diagram).
> > * The core problem is that the authors overpromise. They promise to the readers a simple overview, which clashes with the reality (synthetic data metrics and methods are deeply intertwined, as authors acknowledge often, e.g. in Section 3.6). This clash is also apparent in Figure 1 vs the main structure. Whereas the main structure seems to indicate you can choose your method on what you're primarily interested in (e.g. Figure 3), the actual flow chart (Figure 1) does not reference most of these at all. In the end, there are no methods that really "focus" on improving certain metrics only and explicitly, so the core thesis of the paper is too simplistic.
> > * This quest for simplicity has also led to some categorizations that seem arbitrary and question the reliability of the survey. For example, in the new Figure 3, GOGGLE is given the labels "alignment" and "diversity", yet the GOGGLE paper does not mention diversity or alignment, though it does measure "utility" (a label it doesn't have). Similarly, PATE-GAN makes a big point about utility, yet only has label "privacy", and CTGAN (with labels "utility" and "fidelity") trains a discriminator to "align" synthetic with real samples. The authors acknowledge that Figure 3 shows just the "focus" of the work, but even then it seems many categorizations are not substantiated, which misleads the reader into thinking that the story is simpler than it really is.

---

> ### Author Response · Authors · 2025-12-22
>
> We thank the reviewer for their continued engagement and insightful comments throughout this process. The additional feedback provided has been invaluable, and we believe incorporating these suggestions has directly resulted in an improved version of our survey.
> Below we address the remaining concerns.
>
> > **Feedback 1. (Value for Practitioners):** *Its main claim is that it offers a unique perspective, which it still does not. For example, Jordon et al. and others have similarly structured reviews of synthetic data by "use to practitioners" (my previous point 4, which has not been responded too). Though something like Figure 1 really goes in the direction of added value for practitioners, it could be significantly improved—e.g. more decision functions (e.g. based on a priori knowledge, number of samples, mixed type or not), more granular nodes (there are sometimes still 10 methods to choose from), and "fast inference" and "high-dimensional" is subjective (e.g. please quantify these in the diagram).*
>
>
> **Response 1.:**
> Comparison to prior work: We apologise for overlooking the specific response to the previous point 4 in our last revision. We appreciate this was brought back to our attention.
> We have now updated Section 5 to discuss prior surveys mentioned by the reviewer in their first comment (e.g., Jordon et al.), while also discussing how our survey distinguishes itself from these prior reviews (e.g., Jordon et al. provides a broad review covering the entire field of synthetic data, without focusing in depth on tabular data generation). The changes are in Section 5, in green text.
>
> Granularity in Figure 1: We appreciate the feedback on Figure 1. To address the request for quantification, we now explicitly define *high-dimensional* (>55 features) directly in the figure and caption, as well as *fast inference* (generation time <1s per 10k samples) in the caption of the figure.
> Further, we introduced additional granularity by adding decision nodes for *mixed data types* and *diversity*.
>
>
>
> > **Feedback 2.a. (Overview vs Reality/Figure 1):** *The core problem is that the authors overpromise. They promise to the readers a simple overview, which clashes with the reality (synthetic data metrics and methods are deeply intertwined, as authors acknowledge often, e.g. in Section 3.6). This clash is also apparent in Figure 1 vs the main structure. Whereas the main structure seems to indicate you can choose your method on what you're primarily interested in (e.g. Figure 3), the actual flow chart (Figure 1) does not reference most of these at all.*
>
> **Response 2.a.:**
>
>
> (i) Addressing the structural issue:
> We agree with the reviewer that synthetic data methods and requirements are intertwined, as we discussed in Section 3.6 as well as on page 18.
> Motivated by the feedback we received, in our revised manuscript, we significantly restructured the content to remove the *primary* requirement categorisation. Indeed, we now group methods based on the *subset* of requirements they address. This structure adheres more closely to the complex reality the reviewer points out.
>
>
> (ii) Clarifying the role of Figure 1:
> The figure is intended to serve as a simplified roadmap to guide practitioners, as we have noted also in the paper (on page 3, in orange text), rather than a theoretical taxonomy.
> The branching logic reflects a typical user's decision-making process (e.g., what should we check first?) rather than mutually exclusive theoretical categories.
>
> We rely on Figure 3 (the Venn-Euler diagram) and Table 1 to depict the complex relations between the methods and requirements, which we also support by the discussion on the relationships among requirements in Section 3.6.
>
>
> (iii) The branching logic in Figure 1:
> To ensure Figure 1 remains a useful practical guide without oversimplifying the landscape:
>
> - We deliberately positioned privacy as the first step because, in sensitive settings, the reader should first consider selecting a model based on whether it gives privacy guarantees or not.
>
> - Similarly, we positioned guaranteed alignment as the final step because it can be achieved in modular way by adding constraints-enforcing layers (e.g., the DRL module reviewed in Section 4.5) on top of a model selected in the previous steps of the roadmap.
>
> - We avoided creating separate branches for fidelity and utility precisely because, as reviewer fM1d also notes, they are difficult to disentangle as they tend to reinforce each other.
> We detailed this relation between utility and fidelity in Section 3.6 (as well as the relationships among the other requirements).
>
> We believe this distinction allows Figure 1 to remain an accessible entry point for practitioners, while the restructured text (along with the detailed evaluation metrics for each requirement and the relations between existing methods and the identified requirements, as shown in Figure 3) satisfy the need for a rigorous categorisation.

---

> > ### Author Response · Authors · 2025-12-22
> >
> > > **Feedback 2.b.:** *In the end, there are no methods that really "focus" on improving certain metrics only and explicitly, so the core thesis of the paper is too simplistic.*
> >
> > Response 2.b.:
> > We agree that a rigid categorisation by *primary* requirement can be reductive. As we mentioned above, we have revised the manuscript to remove this primary categorisation in favor of grouping methods by the subset of requirements they address (visualised in the Venn-Euler diagram in Figure 3).
> >
> > However, we maintain that the underlying premise of the paper remains valid. Indeed, while methods often impact multiple metrics (measuring different requirements), it is important to acknowledge that many methods do prioritise specific requirements in their core methodology (often at the expense of others, as detailed in the trade-off analysis in Section 3.6). For example:
> > - CUTS focuses heavily on differential privacy in its design
> > - C-DGMs provide mathematical guarantees for constraint satisfaction (alignment), a feature absent in standard base models.
> >
> > Further, as shown in the C-DGM example, where a module is added on top of an existing model, a method might focus exclusively on one aspect (background knowledge constraints) while inheriting other properties (such as privacy) from the underlying model.
> >
> > These design differences are precisely what motivates this survey.
> > The application landscape for synthetic data is vast and, for example, while a safety-critical application may require certain conditions to be satisfied (e.g., guaranteed alignment), a sensitive healthcare application might prioritise other needs (e.g., privacy). Our survey takes a user-centric perspective and maps these distinct design priorities to user needs, while also advocating for a more rigorous and complete evaluation protocol.
> >
> >
> > > **Feedback 3.a. (Categorisation GOGGLE):** *This quest for simplicity has also led to some categorizations that seem arbitrary and question the reliability of the survey. For example, in the new Figure 3, GOGGLE is given the labels "alignment" and "diversity", yet the GOGGLE paper does not mention diversity or alignment, though it does measure "utility" (a label it doesn't have).*
> >
> >
> > **Response 3.a.:**
> > (i) Alignment: The GOGGLE paper does not explicitly use the specific term *alignment*. However, our survey defines alignment as the capability to embed background knowledge constraints into the models and enforce consistency with these constraints.
> > GOGGLE explicitly addresses this by allowing background knowledge to be injected via an adjacency matrix. We classify this as alignment because the generative process is constrained by this known structural knowledge.
> >
> >
> > (ii) Diversity: Similarly, sampling diversity is helped by intrinsic mechanisms: unlike standard tabular data synthesisers that use fully connected layers where every variable interacts with every other variable, GOGGLE learns a sparse relational structure.
> > This prevents the model from learning ``average'' representations that reduce diversity, which motivated our classification of GOGGLE to the diversity requirement.
> >
> > (iii) Utility: Finally, we actually did assign GOGGLE to the utility requirement in our original submission (GOGGLE indeed heavily emphasises high utility performance). This assignment can be seen in both Table 1 and Figure 3 in our survey.
> > However, we suspect this was missed because the formatting of Table 1 was difficult to scan due to its compactness and lack of separators between the rows in the previous versions of our paper.
> > Following the suggestion of reviewer B4TN, we updated the table layout in the revision (with clear row separators) to ensure these assignments are immediately visible.
> >
> >
> >
> > > **Feedback 3.b. (Categorisation PATE-GAN):** *Similarly, PATE-GAN makes a big point about utility, yet only has label "privacy".*
> >
> > **Response 3.b:**
> > While PATE-GAN shows good utility performance, the breadth of its empirical evaluation is limiting (i.e., the paper compares PATE-GAN almost exclusively against DPGAN). Its defining contribution to the literature is the PATE mechanism for differential privacy. Therefore, its categorisation reflects this distinct focus.

---

> > > ### Author Response · Authors · 2025-12-22
> > >
> > > > **Feedback 3.c. (Categorisation CTGAN):** *CTGAN (with labels "utility" and "fidelity") trains a discriminator to "align" synthetic with real samples.*
> > >
> > > **Response 3.c.:**
> > > The CTGAN paper mentions that the ``discriminator is trained to align synthetic with real samples''.
> > > However, in our survey, alignment strictly refers to the synthetic data's consistency with background knowledge constraints, whereas the term *align* is used informally in the CTGAN paper to refer to the process of matching the real data distribution, falling under the fidelity requirement (which is one of the requirements we assigned to CTGAN in our survey).
> > > Since CTGAN does not provide a mechanism to enforce hard constraints or inject domain knowledge, it does not fit our definition of alignment.
> > >
> > > We agree this difference in terminology could be confusing.
> > > To this end, we added a footnote in Section 2 (page 5) clarifying that while some papers use *alignment* (typically informally) to mean distribution matching, our survey reserves the term for compliance with background knowledge (as formally defined in our Section 3.2).

---

### Review · Reviewer_fM1d · 2025-11-18

**Summary Of Contributions:**

This paper surveys the literature on tabular data generation. The authors identify four key requirements that define high-quality synthetic tabular data: utility (the usefulness of generated data for downstream tasks), alignment (the ability to respect domain constraints and semantics), fidelity (the preservation of statistical properties and complex feature relationships), and privacy (the protection of sensitive information). Based on these dimensions, the paper reviews a wide range of generation techniques and discusses representative methods that are presented as addressing each of these requirements.

**Audience:**

Yes

**Audience Explanation:**

This paper presents a reasonably comprehensive (to the best of my knowledge) overview of existing tabular data generation methods, which should be relevant and of interest to the community. Furthermore, the attempt to outline key requirements for synthetic tabular data and describe how current approaches meet them to varying degrees may help clarify important considerations for future work in tabular data generation.

**Broader Impact Concerns:**

I am not aware of any specific ethical concerns that require more detailed discussion.

**Claims And Evidence:**

No

**Claims Explanation:**

While the paper aims to present a comprehensive survey with a clear categorization of tabular data generation methods, several aspects of the proposed taxonomy are confusing and, in my view, should not be presented in their current form.

For example, in the model types categorization (as illustrated in Figure 1), the authors introduce "layer-based" models as a separate category. However, most models listed under other categories (e.g., GANs, diffusion models) are also inherently layer-based. The resulting classification appears inconsistent and raises questions about the rationale and usefulness of such a distinction.

Similarly, the assignment of methods to the proposed requirement categories (utility, privacy, alignment, and fidelity) is difficult to interpret and risks being misleading. The conceptual relationships among these requirements are not clearly articulated; in practice, some of them tend to reinforce each other (e.g., higher fidelity often supports higher utility), while others are frequently in tension (e.g., the privacy–utility trade-off). Given this, it is unclear how to interpret the categorizations shown in Figure 2. For instance, some methods are presented as satisfying only “privacy”, while others “privacy and utility,” etc., even though any method that claims to be privacy-preserving should, by design, aim to balance privacy and utility at the same time, rather than addressing one in isolation. As currently presented, such categorizations may therefore oversimplify the methodological landscape and obscure rather than clarify the contributions of each approach.

**Requested Changes:**

The core categorization framework of the survey requires substantial restructuring. In its current form, several classification choices are confusing and risk misleading readers. Specifically:

- Model type taxonomy (e.g., the inclusion of layer-based models as a separate category) should be revisited, as this label appears to apply broadly to most architectures included in other categories (such as GANs and diffusion models), making the distinction unclear and of limited value.

- Categorization of methods under the four requirement dimensions (utility, privacy, alignment, fidelity) needs clarification and potential redesign. The conceptual boundaries and relationships among these requirements are not well defined, yet the survey assigns methods to single or combined categories in a way that is difficult to interpret and may oversimplify or misrepresent the inherent trade-offs (e.g., privacy–utility) or natural alignments (e.g., fidelity–utility).

To improve clarity and rigor, I recommend that the authors revise the categorization scheme, provide an explicit justification for category definitions, and ensure that method placement reflects well-differentiated and conceptually well-defined criteria.

---

> ### Author Response · Authors · 2025-12-02
>
> We sincerely thank the reviewer for their insightful comments and constructive feedback. We appreciate your recognition of the survey as relevant to the community and its contribution towards "clarifying important considerations for future work in tabular data generation".
> We agree that the previous categorisation framework (specifically regarding "layer-based" models and the rigid assignment of requirements) was confusing. We have revised the manuscript to incorporate these suggestions (using colors to highlight specific changes).
>
>
>
> > **Feedback 1. (Restructuring):**  *The core categorization framework of the survey requires substantial restructuring. [...]
> The assignment of methods to the proposed requirement categories (utility, privacy, alignment, and fidelity) is difficult to interpret and risks being misleading.*
>
> **Response 1.:**
> We thank the reviewer for their insightful comment, which allowed us to revise and improve the structure of our survey. To this end, we have restructured the survey into "Requirements and Evaluation Metrics" and "Methods" sections. This decoupling allows us to analyse how models address multiple requirements simultaneously, rather than categorising them by a single primary requirement.
> The paper now follows the suggested structure:
> * **Section 3 (Requirements and Evaluation Metrics):** This section details the five requirements (Utility, Alignment, Fidelity, Privacy, and Diversity) and the evaluation protocols and metrics for each of them.
> * **Section 4 (Methods):** This section surveys the methods grouped by their underlying model's architecture (GANs, Diffusion, Transformers, etc.).
>
> This change eliminates the issue of associating models to a single primary requirement. Instead, we now capture the overlap in **Table 1**, which uses a multi-column checkmark system to show that models address multiple requirements simultaneously (and which does not contain the "Primary Requirement" column anymore), as well as in *Figure 3* which visualises the surveyed models by a Venn diagram w.r.t. the requirements.
>
>
>
> > **Feedback 2. (Taxonomy):**  *Model type taxonomy (e.g., the inclusion of layer-based models as a separate category) should be revisited, as this label appears to apply broadly to most architectures included in other categories (such as GANs and diffusion models), making the distinction unclear and of limited value.*
>
> **Response 2.:**
> In order to improve clarity, we revised the taxonomy in **Section 4 (Methods)** (as well as Table 1, which summarises the surveyed methods) such that the methods under the "Layer-based Methods" are now under "Constraints-enforcing Methods".
> Moreover, we clarified on page 2 (in orange) that the decision to enforce domain constraints using one of the "Constraints-enforcing Methods" is positioned as a final step in our newly-added practical roadmap from Figure 1, which serves also as simplified illustration of the survey.
> Indeed, in case domain constraints are needed, the listed methods (i.e., C-DGM and DMG+DRL) to guarantee their satisfaction can be simply built on top of a given synthesiser.
> More precisely, we clarify in the text that these are differentiable layers/modules designed to be integrated on top of any base synthesiser (such as GANs or Diffusion models) during training to guarantee alignment, rather than presenting them as a mutually exclusive model architecture.

---

> > ### Author Response · Authors · 2025-12-02
> >
> > > **Feedback 3. (Relationships among requirements):**  *The conceptual relationships among these requirements are not clearly articulated; in practice, some of them tend to reinforce each other (e.g.,
> > higher fidelity often supports higher utility), while others are frequently in tension (e.g., the privacy–utility trade-off).*
> >
> > **Response 3.:**
> > We have added a dedicated subsection, **Section 3.6 (Relationships Among the Requirements)**, to explicitly discuss these interactions.
> > * We analyse the hierarchy between  fidelity and alignment (noting  (i) that perfect fidelity implies alignment, but not vice-versa, (ii) that alignment is crucial when the real data has errors, and (iii) that methods guaranteeing alignment can be used to check the real data before it is used to train a synthesiser).
> > * We discuss the inherent tension (trade-off) between privacy and utility/fididelity, as well as the complementary relationship between fidelity and diversity.
> > * We discuss the synergy between fidelity and utility.
> >
> >
> > > **Feedback 4. (Regarding Figure 3 interpretation, Venn diagram):**  *it is unclear how to interpret
> > the categorizations shown in Figure 2. For instance, some methods are presented as satisfying only privacy, while others privacy and utility, etc., even
> > though any method that claims to be privacy-preserving should, by design, aim to balance privacy and utility at the same time, rather than addressing one in
> > isolation.*
> >
> > **Response 4.:**
> > In addition to eliminating the methods' grouping by a primary requirement, we have now added a clarification (in purple text) on how to interpret Figure 3 (please note that in the revision the Venn diagram does not correspond to Figure 2 anymore, but to Figure 3) in the context of the requirements, on page 17, under the "Future directions" paragraph.
> > In particular, we state that the categorisation in Figure 3  should be interpreted as a reflection of each method's focus, rather than an exclusive capability.
> > As discussed in Section 3.6, these requirements  are deeply interconnected: they often overlap or trade-off against one another.
> > For example, the methods associated solely with the privacy requirement are positioned as such because their core contribution is a privacy-preserving mechanism.
> > However, this does not implicitly suggest they lack utility (indeed, a synthesiser with very low utility would be useless in any downstream task), but rather that their innovations are  driven by primarily addressing the privacy requirement.
> > Similarly, the methods associated to the fidelity requirement focus on approximating the real data distribution, which often sets the foundation for achieving a high utility.

---

> > > ### Comment · Reviewer_fM1d · 2025-12-24
> > >
> > > I thank the authors for the detailed rebuttal. The manuscript has been significantly improved, and my previous concern regarding the unclear classification of different methods has largely been resolved. Although I remain somewhat conservative about the presentation of Figure 3 (as its message may not be immediately clear to the intended audience of this survey paper), I believe it is acceptable to keep it in its current form.
> > >
> > > I would like to raise one additional minor point. As stated in the abstract, the survey primarily focuses on deep generative models, and the coverage of general tabular data generation methods is therefore not exhaustive. For example, a substantial line of work in the area of private query release generates privacy-compliant tabular “pseudo-data” without relying on explicit generative models. Given this scope, it would be more appropriate to make the focus on deep generative approaches explicit in the title. As it currently stands, the title “A Survey on Tabular Data Generation” may be interpreted as aiming to cover the full landscape of tabular data generation methods.
> > >
> > > Otherwise, I have no further concerns and would recommend acceptance if this point is addressed.

---

> > > > ### Author Response · Authors · 2025-12-24
> > > >
> > > > We thank the reviewer for their positive appraisal and for acknowledging the improvements made during the discussion period. We genuinely appreciate the reviewer's insightful feedback, which we believe has directly led to a significantly stronger manuscript.
> > > >
> > > > To address the final concern, we updated the title to reflect the focus on deep learning methods:
> > > > ``A Survey on Deep Learning Approaches for Tabular Data Generation: Utility, Alignment, Fidelity, Privacy, Diversity, and Beyond''

---

### Decision · Action_Editor_3CtK · 2026-01-07

**Recommendation:** Accept with minor revision

**Additional Comments:**

I am marking this as "accept with minor revision" to allow the authors to address the concern on unjustified claim of providing a "unique" perspective.

**Audience:**

Yes

**Audience Explanation:**

All reviewers agree that there would be some individuals interested in knowing the findings of this paper.

**Claims And Evidence:**

No

**Claims Explanation:**

Two reviewers find the submission to provide sufficient evidence.

Reviewer NnVd submitted their Official Recommendation before the final revisions and at that point answered "no". As far as I can tell, some of their concerns have since been addressed, but one remains:
* Its main claim is that it offers a unique perspective, which it does not. For example, Jordon et al. and others have similarly structured reviews of synthetic data by "use to practitioners". Though something like Figure 1 really goes in the direction of added value for practitioners, it could be significantly improved—e.g. more decision functions (e.g. based on a priori knowledge, number of samples, mixed type or not), more granular nodes (there are sometimes still 10 methods to choose from), and "fast inference" and "high-dimensional" is subjective (e.g. please quantify these in the diagram).

---

> ### Author Response · Authors · 2026-02-10
>
> We sincerely thank the Reviewers and the Action Editor for their feedback and valuable insights, which have led to a stronger version of our paper.
>
> In the camera-ready revision, we removed the phrasing "this survey takes a unique stance", and we added a paragraph in Section 5 (immediately after the Related Work) to highlight our contributions.